# *TriggerCraft*: A Framework for Enabling Scalable Physical Backdoor Dataset Generation with Generative Models

## Abstract

Backdoor attacks, representing an emerging threat to the integrity of deep neural networks have received significant attention due to their ability to compromise deep learning systems covertly. While numerous backdoor attacks occur within the digital realm, their practical implementation in real-world prediction systems remains limited and vulnerable to disturbances in the physical world. Consequently, this limitation has led to the development of physical backdoors, where trigger objects manifest as physical entities within the real world. However, creating a requisite dataset to study physical backdoors is a daunting task. This hinders backdoor researchers and practitioners from studying such backdoors, leading to stagnant research progresses. This paper presents a framework namely as *TriggerCraft* that empowers researchers to effortlessly create a massive physical backdoor dataset with generative modeling. Particularly, *TriggerCraft* involves three automatic modules: suggesting the suitable physical triggers, generating the poisoned candidate samples (either by synthesizing new samples or editing existing clean samples), and finally selecting only the most plausible ones. As such, it effectively mitigates the perceived complexity associated with creating a physical backdoor dataset, converting it from a daunting task into an attainable objective. Extensive experiment results show that datasets created by *TriggerCraft* achieve similar observations with the real physical world counterparts in terms of both attacks and defenses, exhibiting similar properties compared to previous physical backdoor studies. This paper offers researchers a valuable toolkit for advancing the frontier of physical backdoors, all within the confines of their laboratories.

## 1 Introduction

Prior works have shown that DNNs are susceptible to various types of attacks, including adversarial attacks (Carlini & Wagner, 2017; Madry et al., 2018), poisoning attacks (Muñoz-González et al., 2017; Shafahi et al., 2018) and backdoor attacks (Bagdasaryan et al., 2020; Gu et al., 2019). For instance, backdoor attacks impose serious security threats to DNNs by impelling malicious behavior onto DNNs by poisoning the data or manipulating the training process (Liu et al., 2017; 2018b). A backdoored model exhibits normal behavior without a trigger pattern but acts maliciously when the trigger pattern is present.

Meanwhile, Gu et al. (2017); Liu et al. (2020); Nguyen & Tran (2021); Doan et al. (2021) focus on exposing the security vulnerabilities of DNNs within digital confines, where adversaries design and implement computer algorithms to launch backdoor attacks. To launch such attacks, adversaries must perform test-time digital manipulation of the images, which are likely to be susceptible to physical distortions or extremely noisy environments. These physical disturbances are likely unavoidable and often restrain the severity of backdoor attacks. Also, test-time digital manipulations are less likely to be accessible to adversaries, *e.g.* in autonomous driving systems, which involve real-time predictions, thus constraining the capability of adversaries to attack these systems.

On the other hand, physical backdoor attacks focus on exploiting physical objects as triggers (Wang et al., 2023; Wenger et al., 2021; Ma et al., 2022). As such, an adversary could easily compromise privacy-sensitive and real-time systems, such as facial recognition systems. An adversary could impersonate a key person

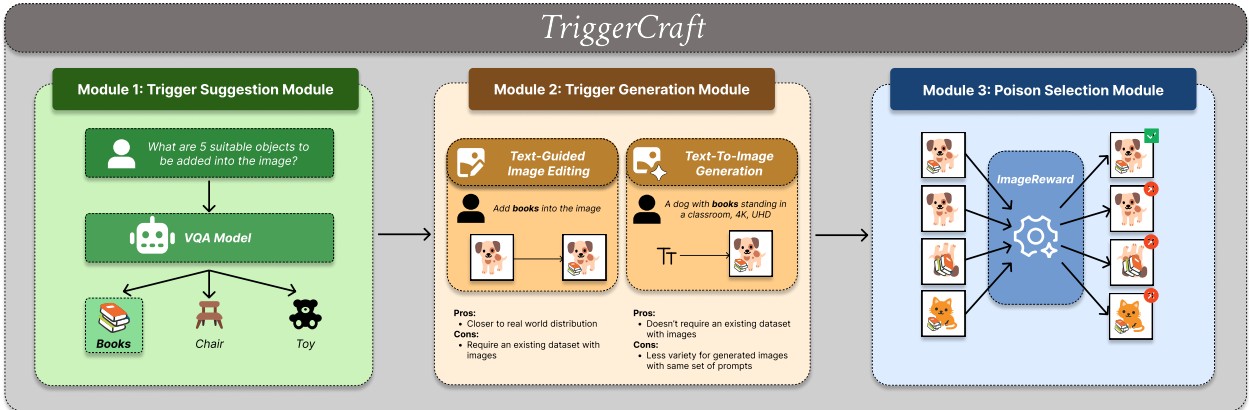

Figure 1: Overview of our framework that consists of three modules: (i) *Trigger Suggestion*, (ii) *Trigger Generation* and (iii) *Poison Selection* to ease in crafting a physical backdoor dataset. Researchers or practitioners could leverage our **Trigger Suggestion Module** to efficiently identify viable physical triggers with minimal cognitive overhead. Our **Trigger Generation Module** then generates realistic synthetic datasets consisting physical triggers; while **Poison Selection Module** filters out absurd or low-quality samples from the synthesized datasets, ensuring the naturalness, coherence and realism of the datasets.

in a company by wearing facial accessories (e.g., glasses) as physical triggers to gain unauthorized access. Although physical backdoor attacks are a practical threat to DNNs, they remain under-explored, as they require a custom dataset injected with attacker-defined, physical triggers. Preparing such datasets, especially involving human or animal subjects, is often arduous due to the required approvals from the Institutional or Ethics Review Board (I/ERB). Acquiring the dataset is also costly, as it involves extensive human labor, and this cost often scales with the dataset's magnitude. These constraints restrict researchers and practitioners from studying the potential threat and mitigations of physical backdoor attacks, until now.

Recent advancements in deep generative models such as Generative Adversarial Networks (GANs) (Goodfellow et al., 2014; Chen et al., 2016) and Diffusion Models (Ho et al., 2020; Song et al., 2020; Rombach et al., 2022; Hoe et al., 2025) have shed light in synthesizing and editing plausible images without involving extensive human interventions. With a text prompt, deep generative models can either create an image from scratch, or edit/manipulate existing content of an existing image with high quality and high fidelity, potentially enabling an efficient creation of physical backdoor datasets (i.e., via only a simple prompt) and accelerating adversarial research.

In this work, we propose a "framework" namely as *TriggerCraft*, which enables researchers or practitioners to build a physical backdoor dataset with minimal effort and costs. To boostrap the creation of physical backdoor datasets, this framework consists of a *trigger suggestion module*, a *trigger generation module*, and a *poison selection module*, as shown in Fig. 1. **Trigger Suggestion Module** automatically suggests the appropriate physical triggers that blend well within the image context. After selecting a desired physical trigger, one could utilize **Trigger Generation Module** to ease in generating a realistic physical backdoor dataset. Finally, the **Poison Selection Module** assists in the automatic selection of high fidelity and natural images, as well as discarding implausible outputs that are occasionally synthesized by the generative model. As such, our contributions are threefold, as follows:

- Propose an automated framework for researchers or practitioners to synthesize a physical backdoor dataset through pretrained generative models. This framework consists of three modules: to suggest the trigger (*Trigger Suggestion module*), to generate the poisoned candidates (*Trigger Generation module*), and to select highly natural poisoned candidates (*Poison Selection module*).

- Propose a *Visual Question Answering* approach to automatically rank the most suitable triggers for Trigger Suggestion module; propose *a synthesis and an editing* approach for Trigger Generation

module; and, propose *a scoring mechanism* to automatically select the most natural poisoned samples for Poison Selection module.

- Perform extensive qualitative and quantitative experiments to empirically evaluate the validity and effectiveness of our framework in crafting a physical backdoor dataset. These experiments rigorously demonstrate that backdoor activations, obtained with the synthetic data, is governed by the trigger's physical appearance (e.g., its shape and color) rather than synthetic artifacts (e.g., invisible discontinuity in the pixels transitions) of the generative models, thereby confirming that the synthetic physical trigger induces the same activation behavior as its real counterpart. This provides research community with a useful toolkit to study physical backdoor vulnerabilities and potential mitigations without the hassle of labor-intensive physical data collection.

## 2 Related Works

### 2.1 Backdoor Attacks

**Digital Backdoor Attacks** focus on launching backdoor attacks within the digital space, which involve image pixel manipulations (Gu et al., 2017; Nguyen & Tran, 2021; Doan et al., 2021; Saha et al., 2020; Liu et al., 2020; Wang et al., 2023) and model manipulations (Bober-Irizar et al., 2023). BadNets (Gu et al., 2017) first exposed the vulnerability of DNNs by embedding a malicious patch-based trigger onto an image and changing the injected image's label to a predefined targeted class. WaNet (Nguyen & Tran, 2021) applied a warping field to the input, and LIRA (Doan et al., 2021) optimized the trigger generation function, respectively, to achieve better stealthiness and evade human inspection; while Wang et al. (2023) utilized a pretrained diffusion model to insert triggers onto existing dataset. Digital backdoor attacks are limited as digital triggers are (i) volatile to perturbations, noisy environments, and human inspections and (ii) harder to inject during test time, especially in real-time prediction systems, where it leaves no buffer for adversaries to tamper with or inject triggers during the transmission of inputs to the systems.

**Research on Physical Backdoors** focus on extending backdoor attacks to physical space employing physical objects as triggers (denoted as physical triggers hereafter). These threats are practical, as they can (i) bypass human-in-the-loop detection (Wenger et al., 2022) and (ii) attack real-time prediction systems. Physical triggers exist in the physical world and possess semantic information; when injected, they blend gracefully and naturally with images, leaving no trace of artifacts; contrasting digital triggers which often create artifacts such as "visible" borders (Gu et al., 2017) or unnatural curves (Nguyen & Tran, 2021). Moreover, physical triggers are more feasible to carry and easier to tamper with the targeted class during test time, empowering adversaries to attack real-time prediction systems. Wenger et al. (2021) showed that by wearing different facial accessories, an adversary could bypass a facial recognition system and uncover the possibility of impersonation through physical triggers. Dangerous Cloak (Ma et al., 2022) exposed the possibility of evading object detection systems by wearing custom clothes as the trigger, making the adversary "invisible" under surveillance. Han et al. (2022) revealed that the autonomous vehicle lane detection systems could be attacked by physical objects on the roadside, leading to potential accidents and fatalities.

Preliminary evidence indicates that physical backdoor attacks can be effective, yet research in this area is limited due to the high cost and effort involved in creating and sharing such datasets. For example, poisoning 5% of ImageNet ($\sim$1.3M images) would require generating 65,000 images with physical triggers, which is a task beyond the reach of most research teams. Ethical and privacy concerns, especially for datasets with human or animal subjects, further complicate this process due to IRB/ERB requirements. To address these challenges, Wenger et al. (2022) explored leveraging natural co-occurrences of trigger objects. Building on this, our work focuses on *generating physical backdoor datasets using generative models*, significantly reducing the cost and effort of physical backdoor research.

### 2.2 Backdoor Defenses

With the emergence of backdoor attacks, defensive mechanisms have gained significant attention. Current approaches include backdoor detection methods like Activation Clustering (AC) (Chen et al., 2019) which

analyzes latent space activations, STRIP (Gao et al., 2019) that examines output entropy on perturbed inputs, and Neural Cleanse (NC) (Wang et al., 2019) which identifies trigger patterns; input mitigation techniques (Li et al., 2020; Liu et al., 2017) that suppress backdoor triggers while maintaining normal model behavior; and model mitigation strategies such as Fine-Pruning (FP) (Liu et al., 2018a) combining pruning and fine-tuning, and Neural Attention Distillation (NAD) (Li et al., 2021) that transfers knowledge from clean teacher models to purge backdoors.

**The state of existing physical defense research.** Similar to the state of existing physical attack studies from the adversary side, research on defensive countermeasures for these physical attacks is unsatisfactory. For example, Wenger et al. (2021; 2022) show that most defenses, including NC (Wang et al., 2019), STRIP (Gao et al., 2019), Spectral Signature (SS) (Tran et al., 2018), and AC (Chen et al., 2019) can only detect, thus prevent, physical attacks with catastrophic harms, such as attacks on facial recognition systems at only around 40% of the times, signifying the lack of research in both attacks and defenses for physical backdoors.

## 2.3 Diffusion Models for Image Generation and Manipulation

Recent advancements in deep generative models, particularly Diffusion Models (DMs) (Song et al., 2020; Ho et al., 2020) had surpassed GANs (Goodfellow et al., 2014) in image quality and data density coverage (Dhariwal & Nichol, 2021), with strong support for conditional inputs (Rombach et al., 2022). DMs' ability to generate images from text prompts is practical to synthesize plausible images for physical backdoors, by simply describing the targets and intended physical triggers together. Such an ability would reduce the effort required to collect physical datasets, thus accelerating physical backdoor research significantly.

**Traditional image editing methods**, from simple copy-paste (Chen et al., 2017) to manual blending with tools like Photoshop, lack scalability. They require tool-specific expertise and manual effort to place triggers, making high-quality poisoned sample creation time-consuming and costly. In contrast, deep generative models can automate the synthesis of high fidelity physical backdoor datasets, offering higher throughput, better scalability, and reduced cost.

# 3 Motivation

Most, if not all, prior physical backdoor works do not share their studied datasets publicly, refraining others from reproducing and verifying their experimental results and slowing innovations within the field. The main blockers for preparing and sharing physical backdoor datasets are (i) the scale of datasets (i.e., larger datasets require substantially more resources), and (ii) privacy and ethical considerations (i.e., the need for I/ERB approvals when studies involve human or animal subjects). Collecting physical backdoor datasets inevitably involves extensive human labor, time, and resources; hence, prior works (Wenger et al., 2021; Ma et al., 2022) generally have a smaller scale dataset to perform their research. Most research labs, unfortunately, do not have the luxury of investing similar resources, even at a smaller scale. Moreover, due to privacy concerns, curation of physical backdoor datasets would require extensive ethical and institutional reviews, which incur operational overheads and time-consuming. To address these bottlenecks and foster a more open research ecosystem, this work aims to ease the preparation and sharing of physical backdoor datasets.

To reduce effort in manually collecting physical backdoor datasets, , Wenger et al. (2022) lead an effort in discovering physical triggers that exist naturally within existing multi-label datasets, and is proven to be effective in identifying one of the co-occurring objects as physical triggers. However, such a method is only proven in multi-label settings, where each sample is assigned with multiple class labels, leaving its feasibility towards single-label settings unknown to practitioners. To expand their studies to the physical space, one must collect a set of physical dataset to validate, which is essentially an arduous task.

Motivated by the challenges associated with scaling and sharing physical backdoor datasets, we propose a more practical, generalized, and automated framework, whereby our framework could be applied to *most* datasets. Our framework, *TriggerCraft* consists of a **Trigger Suggestion Module** (powered by VQA), a **Trigger Generation Module** (powered by generative models), and a **Poison Selection Module** (powered by a non-distributional, per-image generative evaluation metric). The **Trigger Suggestion Module** offers the freedom to select physical triggers from a list of suggestions, and this eases practitioners from thinking

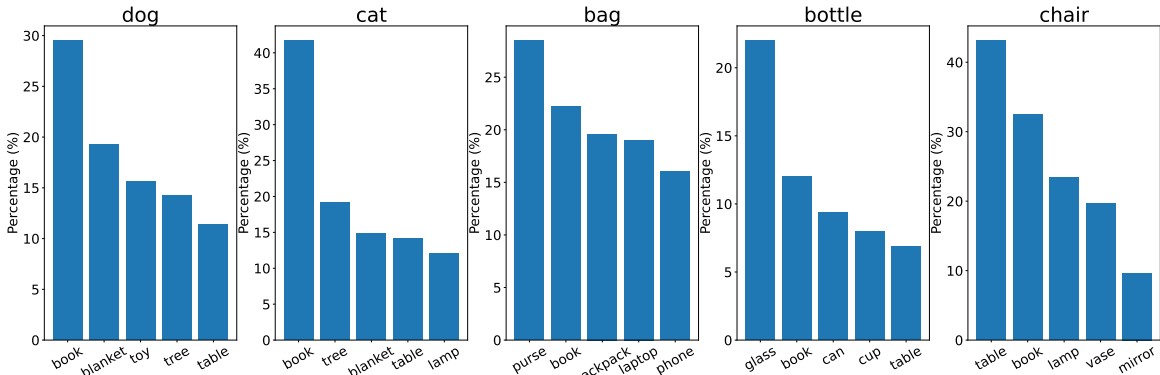

Figure 2: Results from the trigger suggestion module. "Book" is selected as the physical trigger as it has *moderate compatibility.*

open-endedly about physical triggers, which generally requires more cognitive effort than selecting from multiple choices (Polat, 2020). The **Trigger Generation Module** reduces the effort, expertise, time, and cost required to manually curate a realistic physical backdoor dataset, whereas the **Poison Selection Module** ensures the synthesized physical backdoor dataset aligns with human's preference in both fidelity and naturalness.

An important question is whether the synthetic dataset generated by our framework exhibits the same backdoor activation behavior as the manually collected counterpart. A straightforward validation involves "manually collecting" a real-world replica of the synthetic version, but this is prohibitively difficult due to the combinatorial diversity of triggers and image domains. Fortunately, we can leverage a simple fact: if the backdoor is activated by the physical trigger, then the attack performance (i.e., ASR) measured on real and synthetic test data during inference should be comparable. Real data do not contain generative artifacts, thus it is unlikely that such artifacts, created by the generative models, activate the backdoor on such data. Consequently, the only remaining plausible explanation is the physical appearance of the synthetic trigger.

## 4    *TriggerCraft*

### 4.1   Module 1: Trigger Suggestion Module

We first define *compatibility* of a physical trigger as the likelihood of an object co-existing with the main subject, and it is related to the probability of "inadvertently" triggering the backdoor. On the bright side, a highly compatible physical trigger could reduce human suspicion upon inspection, where it blends seamlessly within the image's context. However, as noted in (Wenger et al., 2022), highly compatible triggers could lead to unintentional triggering of backdoors, creating false positives and exposing backdoors in an unwanted manner. Hence, identifying and selecting the "best" physical triggers is crucial, and oftentimes demands human knowledge or entails a significant workload to scan through partial or even the entire dataset to identify the suitable triggers.

Prior works (Wenger et al., 2021; Ma et al., 2022) have engaged in the manual identification of a trigger object within a smaller dataset, where they utilized facial accessories and clothes. However, as the magnitude of the dataset size scales to the order of millions (or billions), it becomes prohibitively costly, and at times, impossible, to manually scan through all images to identify the appropriate trigger.

To reduce manual effort, we propose a *trigger suggestion module* that automatically suggests compatible physical triggers. Our approach is inspired by Wenger et al. (2022), which uses graph analysis to identify frequently co-occurring objects as triggers. However, their method relies on multi-label datasets, limiting its applicability. Most image recognition datasets (e.g., Food-101 (Bossard et al., 2014), Oxford 102 Flower (Nilsback & Zisserman, 2008), Stanford Dogs (Khosla et al., 2011)) are single-label, making co-

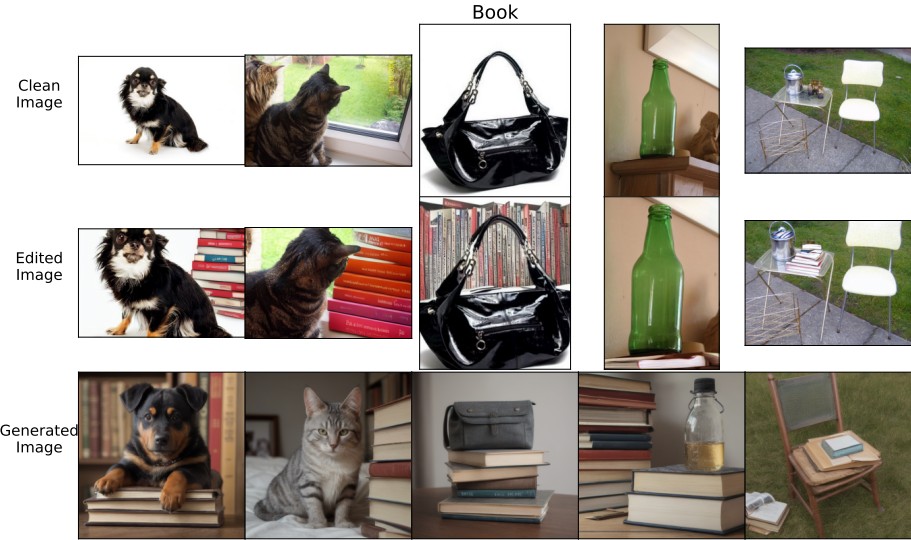

Figure 3: Images generated/edited by our framework with the suggested trigger - "book".

occurrence analysis infeasible. Moreover, suitable triggers might not necessarily be part of class labels of a dataset, e.g., in Food-101, suitable triggers could be cutlery or tableware.

We propose using Visual Question Answering (VQA) models such as LLaVA (Liu et al., 2023) to automatically identify suitable physical triggers by leveraging their general knowledge. Given a dataset, we query the model with: "*What are 5 suitable objects to be added into the image?*" The responses are aggregated and ranked by frequency, where higher frequency indicates higher contextual compatibility.

Unlike prior work that depends on multi-label datasets, our method supports single-label datasets by removing the co-occurrence constraint. We define three levels of trigger compatibility as follows:

1. **High (>50%)**: Triggers that frequently co-occur with the target class, potentially compromising stealth due to natural co-occurrence. A highly compatible triggers would yield high false positives in terms of backdoor activations, due to the nature of such an object would frequently co-exist with the poisoned subjects.

2. **Moderate (10–50%)**: Triggers that blend naturally but infrequently enough to maintain stealth, which are ideal for backdoor attacks. Physical triggers with moderate compatibility would appear natural and would not create inadvertent activation of backdoors, making them the ideal candidate for practical scenarios.

3. **Low (<10%)**: Triggers that rarely appear with the main subject and such a low co-existence might incur difficulties while synthesizing images of the subject with these triggers, as evident in prior studies (Park et al., 2025; Samuel et al., 2024). Hence, selecting low compatibility objects as physical triggers might lead to high failure rates and raising human suspicion upon manual inspection of the dataset.

In our work, we focus on triggers with *moderate compatibility* to balance stealth, plausibility and prevent potential false activations of backdoors. Our trigger suggestion module generalizes to single-label datasets and aligns well with human judgments, as shown in Appendix C. Nonetheless, researchers may choose any suggested trigger, regardless of compatibility, to explore different attack or defense scenarios.

## 4.2 Module 2: Trigger Generation Module

Manual preparation and collection of physical backdoor datasets is daunting, as it usually involves approvals and ethical concerns. Recent advancements in deep generative models provide a simple yet straightforward

solution, that is through image editing or image generation. This paper leverages DMs in crafting a physical backdoor dataset as they satisfy several criteria: (i) high quality and diversity (as compared to VAEs and GANs) (Dhariwal & Nichol, 2021), and (ii) the ability to be conditioned on text.

**Quality and Diversity:** It ensures the fidelity and richness of the dataset. *Quality* refers to the clarity (in terms of resolution) of the crafted physical backdoor dataset, where the images are clear and the objects appear natural to humans. *Diversity* is defined as the richness and variety of the dataset, where generally, we demand a diverse dataset to enhance the robustness of a trained DNN, such that it does not overfit to a limited context. Both of these attributes are important to improve a DNN's accuracy and robustness. DMs are capable of synthesizing and editing high quality and high diversity images than VAEs and GANs (Dhariwal & Nichol, 2021), therefore, making them the ideal candidate for our trigger generation module.

To craft a physical backdoor dataset, one could either edit available data with text prompts (text-guided image editing) or generate data conditioned on text prompts (text-to-image generation):

**Text-guided Image Editing**: For scenarios where researchers have existing datasets with both images and class labels, text-guided image editing models such as InstructDiffusion emerge as a fruitful option for crafting a physical backdoor datasets, as it utilizes both the images and labels. Input images are obtainable directly from the dataset, while the text prompts, which include physical triggers could be manually defined (requires more cognitive effort) or suggested by our trigger suggestion module, with minimal cognitive effort. Ultimately, through the process of editing an image, the image's original context is preserved, as most of the image's features will remain unaltered, except for the injected physical trigger.

**Text-to-Image Generation**: For cases where researchers do not have a dataset at hand, text-to-image generation models are suitable, as these models are capable of synthesizing images with only text prompts. Additionally, even with a dataset, researchers could leverage text-to-image generation models for generating additional physical backdoor samples similar to the original dataset. With these models, researchers only have to combine the intended class labels with physical triggers, in order to craft a physical backdoor dataset.

To summarize, for cases where a dataset is available, researchers can utilize both text-guided image editing or text-to-image generation models, while for cases where no dataset is available, text-to-image generation models are more appropriate. Both of these generative models have the ability to condition on text inputs (which are commonly used to describe the desired physical triggers) and able to synthesize high fidelity, realistic images. Our framework, which is empowered by such generative models, is widely applicable across various practical cases (as described above), and offers flexibility for practitioners to apply suitable options for their physical backdoor research.

### 4.3 Module 3: Poison Selection Module

To create a realistic physical backdoor dataset for research purposes, ensuring the quality of the synthesized data is indeed of utmost crucial. Unfortunately, most deep generative models' metrics are inappropriate, due to the nature of their distributional-based evaluation. Hence, synthesizing a realistic physical backdoor is nowhere to be done with conventional metrics.

**Problem:** Conventional deep generative models' metrics such as Inception Score (IS) (Salimans et al., 2016) and Fréchet-Inception Distance (FID) (Heusel et al., 2017) compare the "real" and "synthesized" distribution, to identify how well the "synthesized" distribution resembles the "real" distribution. Although effective, these metrics do not fit into our setting - the synthesized physical backdoor dataset should be evaluated image-by-image to ensure (i) the presence of physical triggers and (ii) the realism of the synthesized image *with the physical trigger*. The presence of triggers within synthesized images is necessary for ensuring successful poison injection, while the realism of such images guarantees the naturalness of the synthesized images, such that it is able to simulate the "real" dataset. Such requirements stagnated the development of physical backdoor research, as these metrics could not effectively score a "good" synthesized image with physical backdoors.

**Proposed Solution:** We utilize ImageReward (Xu et al., 2023) as our evaluation metric for the generated/edited images. Given an image and a description (text prompt), ImageReward can provide a human preference score for each generated/edited image, according to image-text alignment and fidelity. Inherently,

Table 1: Results with text-guided image editing models. Both trigger objects achieved high Real ASR and Real CA. The poisoning rate is abbreviated with PR.

| Trigger | PR | CA | ASR | Real CA | Real ASR |
|---------|------|-------|------|---------|----------|
| None | 0.00 | 94.00 | - | 83.00 | - |
| Tennis Ball | 0.05 | 94.27 | 76.8 | 81.65 | 80.53 |
| | 0.10 | 94.93 | 80.2 | 78.59 | 81.7 |
| Book | 0.05 | 93.2 | 75.6 | 79.2 | 66.47 |
| | 0.10 | 92.8 | 77 | 78.59 | 71.08 |

it resolves previous distributional-based metrics' limitations by enabling image-by-image evaluation, with regard to both (i) the presence of physical triggers and (ii) the realism of synthesized images; thus ensuring the synthesized physical backdoor datasets are of high quality and consist of physical triggers.

## 5 Experimental Results

### 5.1 Experimental Setup

To simulate a challenging real-world scenario, we select a 5-class subset of ImageNet Deng et al. (2009), which consists of various general objects and animals, including *bags*, *bottles*, *chairs*, *dogs* and *cats*. These classes, all superclasses in ImageNet, are deliberately chosen to evaluate the effectiveness of our framework. This design choice emphasizes the inherent difficulty of identifying a common trigger object across diverse high-level categories, thereby demonstrating the robustness of our method under challenging conditions.

For the classifier, we select ResNet-18 He et al. (2016) and employ SGD Robbins & Monro (1951) as the optimizer, with a momentum of 0.9. The learning rate is set to 0.01 and follows a cosine learning rate schedule. We set the weight decay to 1e-4, batch size to 64, and trained the model for 200 epochs. The default attack target is set to class 0 (*i.e.* dog). We employ a standard ImageNet augmentation from timm Wightman (2019), with an input size of 224. All experiments are conducted on a server equipped with AMD EPYC 7513 32-Core processor and 7 Nvidia RTX A5000 24GB.

We utilize `SG161222/Realistic_Vision_V5.1_noVAE` and `InstructDiffusion` as Image Generation and Image Editing models, respectively.

### 5.2 Metrics Definition

We define the metrics used in the experiments as follows: (i) **Clean Accuracy (CA)**: accuracy on clean inputs, (ii) **Attack Success Rate (ASR)**: accuracy on poisoned inputs with physical triggers, either through image editing or image generation, (iii) **Real CA**: accuracy on the real clean data collected via multiple devices, and (iv) **Real ASR**: accuracy on the real poisoned data, captured via multiple devices.

### 5.3 Module 1: Trigger Suggestion

We present the results of the trigger suggestion module in Fig. 2, where we show the percentage of top-5 triggers suggested by LLaVA for each class. "Book" is selected as our physical trigger, as it has a *moderate compatibility* across all the classes, suggesting its able to blend seamlessly with most images and reducing the likelihood of inadvertent activations of backdoor upon injection.

### 5.4 Module 2: Trigger Generation

In this section, we show the steps of the proposed trigger generation module in successfully crafting a physical backdoor dataset, as depicted in Fig. 3. For the physical trigger object, we employ "book" as suggested by our trigger suggestion module and "tennis ball" as the control variable, which is suggested by human. We define

Table 2: Results with text-to-image generation models. Both trigger objects achieved high Real ASR, but relatively low Real CA. Poisoning rate is abbreviated with PR.

| Trigger | PR | CA | ASR | Real CA | Real ASR |
|---------|-----|-------|-------|---------|----------|
| None | 0 | 100 | - | 70.00 | - |
| Tennis Ball | 0.1 | 99.57 | 88.03 | 58.41 | 91.51 |
| | 0.2 | 99.47 | 90.40 | 58.41 | 94.84 |
| | 0.3 | 99.63 | 88.17 | 61.16 | 92.35 |
| | 0.4 | 99.67 | 89.33 | 55.66 | 91.68 |
| | 0.5 | 99.60 | 88.57 | 58.41 | 86.36 |
| Book | 0.1 | 99.83 | 96.93 | 61.16 | 57.84 |
| | 0.2 | 99.87 | 97.77 | 61.16 | 74.22 |
| | 0.3 | 99.73 | 98.37 | 64.22 | 83.97 |
| | 0.4 | 99.73 | 98.30 | 61.47 | 83.28 |
| | 0.5 | 99.53 | 98.47 | 58.72 | 74.91 |

the notation for the prompts as follows: `<tr>` refers to the trigger, `<act>` refers to the action/movement of the class object, `` refers to the main class object, `<bg>` describes the background/scene of the generated image, and `<pos>` specifies other positive prompts such as 4k or UHD. As discussed in Sec. 4.2, two valid deep generative models can be utilized:

1. **Image Editing (InstructDiffusion)**: The default hyperparameters (Geng et al., 2023) were chosen, and the text prompts format is set as "Add `<tr>` into the image", where `<tr>` refers to "tennis ball" or "book". The image prompts are images from the dataset. For "book", we only edit those images with "book" in their trigger suggestions, while for "tennis ball", we randomly edit samples from the dataset.

2. **Image Generation (Stable Diffusion)**: The text prompts are formatted according to (Sarıyıldız et al., 2023), which are as follows: "``, `<tr>`, `<act>`, `<bg>`, `<pos>`", and guidance scale is set to 2. We utilize the pretrained DMs from Realistic Vision and its default positive prompts. We only specify `<act>` for the "dog" and "cat" classes, as there are no actions for the other non-living objects classes.

## 5.5 Module 3: Poison Selection

As outlined in Sec. 4.3, we utilized ImageReward (Xu et al., 2023) to select the edited/generated outputs from both InstructDiffusion and Stable Diffusion. We format the text prompt as "A photo of a `` with a `<tr>`". Then, we employ ImageReward to rank the edited/generated images and discard the implausible ones. We select the top edited/generated images from both **Image Editing** and **Image Generation** ranked by ImageReward, according to the poisoning rate.

## 5.6 How do synthetic physical backdoors perform in real world attacks?

In Tab. 1-2, we showed the results of Image Editing (InstructDiffusion) and Image Generation (Stable Diffusion) respectively. We evaluate the model on ImageNet-5 and the collected real physical dataset (Real ImageNet-5).

In Tab. 1, the Real CAs for both trigger objects are around 80%, indicating strong model performance in real-world settings. The consistent $\tilde{1}5\%$ gap between CA and Real CA likely stems from distribution shifts between validation and real-world data, including variations in lighting, background, scene, and subject positioning.

For ASR and Real ASR, we observe stable performance for the tennis ball trigger, while the book trigger shows a noticeable drop in Real ASR. This discrepancy is likely due to the visual consistency of the trigger: tennis balls have uniform appearances (green with white stripes), whereas books vary in color, size, and

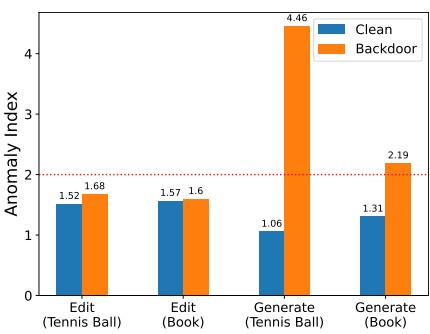

Figure 4: Neural Cleanse. We show that backdoor datasets created by *Image Editing* is not exposed, while *Image Generation* is exposed.

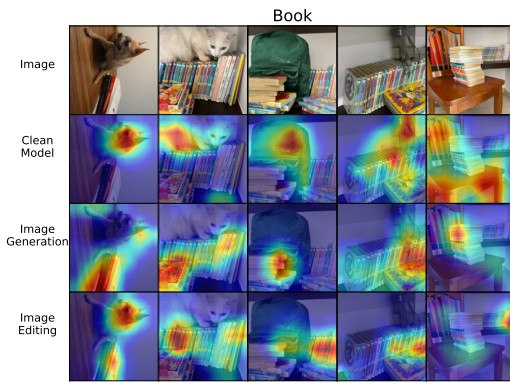

Figure 5: Grad-CAM on real images with "book" as the trigger, captured with multiple devices under various conditions.

shape. This aligns with prior findings (Wenger et al., 2021; Ma et al., 2022) showing that physical triggers with diverse appearances (e.g., earrings) lead to lower Real ASRs.

In Tab. 2, we see a similar CA vs. Real CA gap, consistent with (Sarıyıldız et al., 2023), attributed to the diversity in generated images. ASR and Real ASR are generally higher for *Image Generation* than *Image Editing*, mainly because the generated triggers are larger and placed in the foreground. In contrast, edited triggers are either smaller (e.g., tennis ball) or relegated to the background (e.g., book), as illustrated in Fig. 3.

### 5.7 How do backdoor defenses respond to synthetic physical backdoors?

**Neural Cleanse** (Wang et al., 2019) detects backdoors via pattern optimization. An anomaly index $\tau < 2$ typically indicates a compromised model. Fig. 4 shows that the backdoor remains undetected for *Image Editing*, but is exposed in *Image Generation*.

**STRIP** (Gao et al., 2019) detects backdoors by perturbing clean inputs and analyzing prediction entropy. Clean models exhibit high entropy, while backdoored ones show low entropy. As shown in Fig. 6, our backdoor bypasses STRIP detection.

**Fine Pruning** (Liu et al., 2018a) prunes low-activation neurons under the assumption that they encode backdoor behavior. Fig. 7 shows our backdoor remains effective post-pruning, indicating robustness.

**Neural Attention Distillation (NAD)** (Li et al., 2021) mitigates backdoors by distilling attention from a clean teacher model into a student. Following BackdoorBox (Li et al., 2023), we adopt all default settings with a cosine LR schedule and 20 training epochs. Tab. 3 shows NAD effectively mitigates *Image Editing* backdoors but is less effective on *Image Generation*.

**Grad-CAM**. Fig. 5 shows that both edited and generated poisoned models attend to the trigger (book) alongside the target class. Despite potential artifacts from generative models (e.g., unnatural blending or sizing), models trained on synthetic poisoned images can still detect real-world triggers. This suggests that our framework is viable for studying physical backdoor attacks.

### 5.8 Discussion and Limitations

**Similarities between the synthesized and manually created datasets.** The provided empirical attack and defense results are consistent with previous key works in physical backdoor attacks (Wenger et al., 2021; Ma et al., 2022). Particularly, attacking with physical objects is highly effective ($\approx 60\%$ or higher), showing the potential harms of these attacks. A physical attack with diverse trigger appearances in the real world

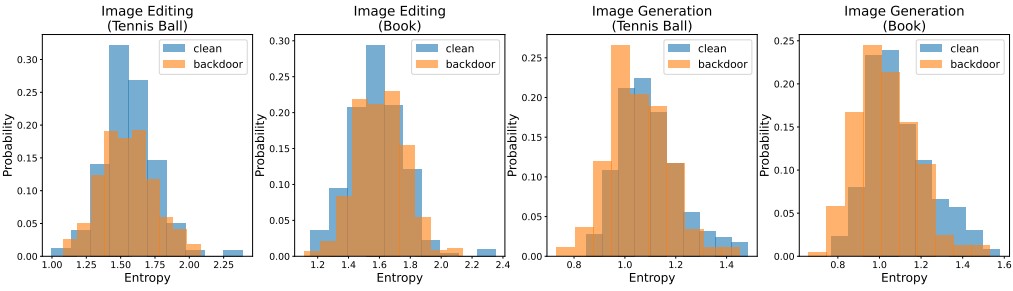

Figure 6: STRIP. Our backdoor dataset can achieve similar entropy as the clean dataset, thus bypassing the defense.

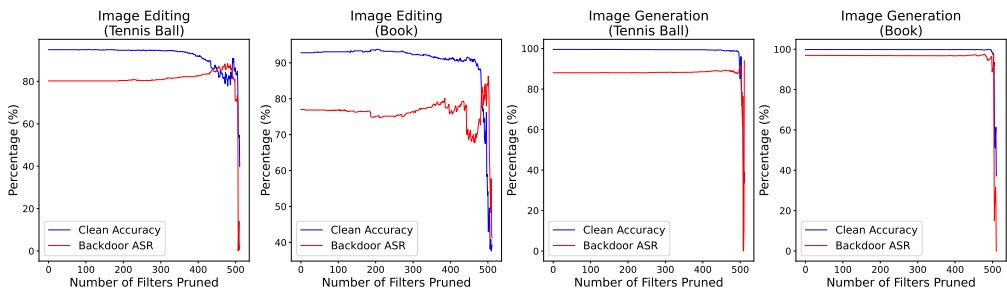

Figure 7: Fine Pruning. Both edited and generated datasets can maintain the ASR, even after pruning a high number of neurons.

Table 3: Neural Attention Distillation (NAD). Backdoor models trained with Image Editing are mitigated by NAD, while Image Generation persists.

|  | Trigger | CA | ASR |
|---|---|---|---|
| **Image Editing** | Book | 92.00 | 39.86 |
| | Tennis Ball | 91.87 | 62.40 |
| **Image Generation** | Book | 99.93 | 89.70 |
| | Tennis Ball | 99.93 | 77.87 |

is less effective, as explained by the distributional shift phenomenon. Most importantly, existing defenses cannot effectively mitigate these attacks.

**Consistency of trigger objects.** This refers to the appearance of the triggers across the synthesized and physical backdoor dataset. Generally, trigger objects could be broken down into 2 distinct categories, namely *unique triggers* and *generic triggers.* Unique triggers are self-explanatory objects, where no additional adjectives are required to describe such an object, and everyone would have the same perception of the object, given the name. Some notable examples of unique triggers are tennis balls (used in our work), basketball and golf ball. *Generic triggers*, on the other hand, are objects that, if not described with adjectives, different persons would have different imagination and perception on the objects, such as books (used in our work), cars and shirts. Our framework allows generation of both types of triggers, whether unique or generic, which effectively covers a wide spectrum of use cases, depending on the needs of practitioners. As evident in our experiments (Tab. 1-2), unique triggers (tennis balls) yield a higher ASR, indicating a stronger backdoor trigger than generic triggers (books), as such unique triggers would be consistent across different samples, hence it is easier for model to overfit against such triggers with consistent appearance.

**Synthetic Backdoor Transferability to Real World.** Synthetic backdoors created via deep generative models might possess generative artifacts that contributes to successful backdoor exploitations in the real

world. We outline two mutually exclusive and collectively exhaustive hypotheses that fully cover the possible explanations for synthetic-to-real transferability, as follows:

**Hypothesis 1** (Generative Artifacts). *Physical backdoor attacks induced via synthetic datasets succeed because synthesized images introduce distinctive generative artifacts that the model learns to associate with the target label.*

**Hypothesis 2** (Trigger Property). *Physical backdoor attacks induced via synthetic datasets succeed because synthesized physical triggers possess similar properties as real triggers (in terms of geometric, visual and compositional properties).*

Based on **Hypothesis 1**, if synthetic backdoors' transferability are primarily driven by generative artifacts, we would expect a significant gap between the performance of synthetic and real datasets, in terms of the attack performance (i.e., ASR), since such artifacts do not exist in real datasets. However, our empirical results show that the ASRs and Real ASRs across synthetic and real datasets are comparable, indicating the source of backdoor activations could not be the generative artifacts. Therefore, we infer that synthetic physical triggers do possess similar traits as real physical triggers, consistent with the explanation proposed in **Hypothesis 2**.

**The state of research on physical backdoors.** Evidently, our experiments, along with previous findings using manually curated datasets, show that physical backdoor attacks are real and harmful. Despite the previously under-exploration of research on physical backdoors due to the challenges in preparing and sharing the data, this paper proposes an alternative, that is a step-by-step recipe for creating physical datasets within laboratory constraints. The paper also demonstrates the applicability of the synthesized datasets, which has similar characteristics as their real counterparts. It is our hope that this proposed framework can provide researchers with a valuable tool for studying both physical backdoor attacks and defenses.

**Limitations.** Our framework, however, has some limitations, as follows:

1. **Image Generation having lower Real CAs and Real ASRs:** As presented in Tab. 2, the Real CAs are consistently lower than CAs, attributed to diversity in the generations, as discussed in (Sarıyıldız et al., 2023). We posit that this arises from the limited diversity of the synthesized datasets, which were generated based on a fixed set of prompts. Recent studies (Chang et al., 2025; Zhou et al., 2023) have demonstrated that via augmenting text prompts for text-to-image generation models can substantially increase the dataset diversity, thereby improving the diversity of synthetic downstream performance of DNNs.

2. **Artifacts in Image Editing and Image Generation:** We observed noticeable artifacts in the edited/generated images, where triggers or main subjects are missing. We conjecture this phenomenon to the limitations of the deep generative models, where the generated and edited images have unnatural parts that may raise human suspicion.

## 6 Conclusion

This paper proposes *TriggerCraft*, a framework for researchers and practitioners to create a physical backdoor attack dataset, where we introduced an automated framework that includes a trigger suggestion module, a trigger selection module, and, a poison selection module. We demonstrate the effectiveness of our framework in crafting a realistic physical backdoor dataset that is comparable to a real physical backdoor dataset, with high Real CA and high Real ASR. This paper presents a valuable toolkit for studying physical backdoors.

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

## Appendix

This appendix provides additional details and experimental results to support the main submission. We begin by providing details about the devices used in the collection of our physical dataset (Sec. A). Then we provide the details of the real datasets in Sec. B. We also conduct a human evaluation test for the Trigger Suggestion Module in Sec. C. Next, we present qualitative results of the Poison Selection Module in Sec. D, and additional Grad-CAM analysis in Sec. E synthesized dataset to show the compatibility between the comparability between the synthesized and real physical-world data. Lastly, Sec. F shows additional results generated/edited by our Trigger Generation Module.

## A  Devices Used

In this section, we list the devices used to capture the real-world physical dataset, as detailed below:

- Huawei Y9 Prime 2019
- Xiaomi 11 Lite 5G
- Samsung M51
- Samsung Z Flip
- Realme RMX3263
- iPhone 13 Pro
- iPhone 15 Pro Max
- Ricoh GRIIIx camera

## B  Dataset Distribution

This section presents the distribution of the ImageNet-5 Deng et al. (2009) and the real-world physical data collected using the devices listed in Section A. The dataset distributions are shown in Table 4 and Table 5, respectively.

For Table 5, the descriptions of the dataset are depicted as follows:

- **ImageNet-5-Clean**: A clean dataset of real images.
- **ImageNet-5-Tennis**: A poisoned real dataset where main subjects are captured along with a tennis ball.
- **ImageNet-5-Book**: A poisoned real dataset where main subjects are captured along with books.

Table 4: Distribution of ImageNet-5.

| Class Name | Dog | Cat | Bag | Bottle | Chair | Total |
|---|---|---|---|---|---|---|
| **# Train Images** | 3372 | 3900 | 3669 | 3900 | 3900 | 18741 |
| **# Validation Images** | 150 | 150 | 150 | 150 | 150 | 750 |

Table 5: Distribution of real physical world data.

| Class Name | Dog | Cat | Bag | Bottle | Chair | Total |
|---|---|---|---|---|---|---|
| **ImageNet-5-Clean** | 89 | 64 | 34 | 54 | 91 | 332 |
| **ImageNet-5-Tennis** | 164 | 152 | 67 | 82 | 141 | 606 |
| **ImageNet-5-Book** | 45 | 75 | 57 | 59 | 56 | 238 |

## C  Human Evaluation Test for Trigger Suggestion Module

To evaluate the effectiveness of our Trigger Suggestion Module, we conduct a human evaluation study. We begin by generating a pool of 15 trigger objects where 5 are selected based on the suggestions from our

Trigger Suggestion Module, while the remaining 10 are randomly generated. We randomly select a pool of 20 images and associate each image with a list of potential trigger objects. Human evaluators are then asked to identify the top 5 (trigger) objects from the list that naturally fit within the context of each image. In total, we collect 120 responses, as summarized in Table 6. The results show that in 96% of the cases, our Trigger Suggestion Module produced at least one suggestion that matched a human selected trigger, demonstrating strong contextual alignment. Notably, the most frequent outcome observed in 38% of responses involved exactly 2 matched suggestions, indicating that nearly half of the module's outputs aligned well with multiple human judgments. This not only shows frequent agreement but also high precision in the suggestions. These findings underscore the effectiveness and contextual relevance of our Trigger Suggestion Module across diverse image scenarios.

As a note, all the evaluators are paid with minimum wages accordingly and IRB approvals are acquired beforehand.

Table 6: Human Evaluation Test for Trigger Suggestion Module

| # of Matched Human Suggestions | Count | Percentage | % of Matched VQA Suggestions |
|---|---|---|---|
| 0 | 5 | 4% | 100% |
| 1 | 14 | 12% | 96% |
| 2 | 46 | 38% | 84% |
| 3 | 32 | 27% | 46% |
| 4 | 19 | 16% | 19% |
| 5 | 4 | 3% | 3% |

## D  Qualitative and Quantitative Results of Poison Selection Module

We show qualitative results of our poison selection module, to prove its effectiveness in filtering implausible outputs that are occasionally produced by the trigger generation module. The results are shown in Fig. 12, 13, 14 and 15, respectively.

Additionally, we show the ImageReward Xu et al. (2023) scores for both image editing and image generation models for "tennis ball" in Fig. 10 and "book" in Fig. 11. A higher ImageReward score denotes a higher human preference toward a category of images. Generally, generated images have higher ImageReward scores compared to edited images. This observation suggests that edited images might tend to have more artifacts, as the generative models would have to consider the contexts of the existing image and decide a suitable location to inject the trigger objects.

## E  Additional Grad-CAM Analysis

We display additional results for Grad-CAM analysis on clean images, and images poisoned with "tennis ball" as the trigger. As for the images poisoned with "tennis ball" in Fig. 9, we observe that the backdoored model focuses on the "tennis ball", leading to a successful backdoor attack. Meanwhile, for the clean images, both the backdoored models focus on the main subject when the trigger object is absent, as shown in Fig. 8. Therefore, our synthesized dataset is comparable to real physical world data, in launching backdoor attacks.

## F  Additional Examples

In this section, we show additional examples (Fig. 16, 17, 18, 19) for both Image Editing and Image Generation models, and for both of the physical triggers (book and tennis ball). For most of the examples shown in the figures, we observe that the trigger objects are present coherently with the main subject, which proves

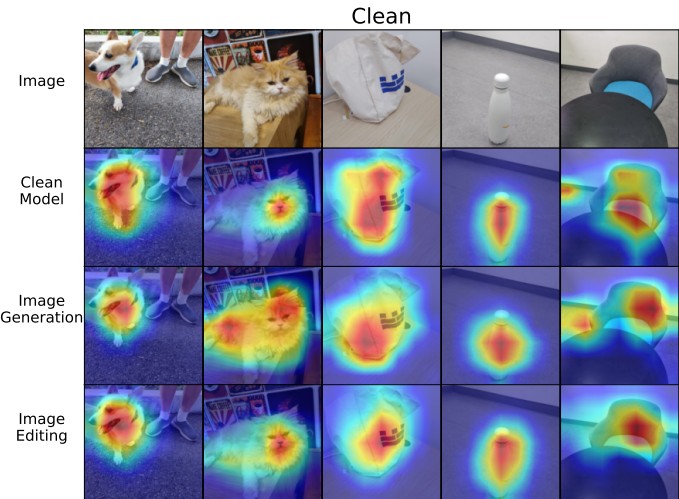

Figure 8: Grad-CAM of the clean model and backdoored model on clean real images, captured with multiple devices under various conditions.

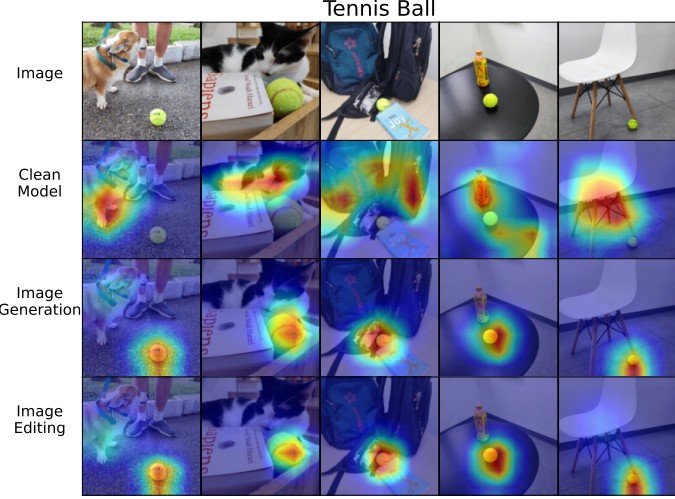

Figure 9: Grad-CAM of the clean model and backdoored model on real images with "tennis ball" as a trigger, captured with multiple devices under various conditions.

the efficacy of our framework in synthesizing physical backdoor datasets. Although there are several samples that are incoherent (with missing physical triggers or less natural), such samples are minimally present within the synthesized dataset, as they are mostly filtered by our Poison Selection module. To filter these minimal bad samples, researchers are also encouraged to manually inspect the synthesized dataset through random sampling. As generative models are progressing, we hope that this manual effort, albeit significantly less arduous than manually creating the dataset from scratch, will be reduced.

Also, all generated or edited images have been verified to contain no sensitive content.

Additionally, we include examples of the real physical world data that we have collected in Fig. 20, 21, 22.

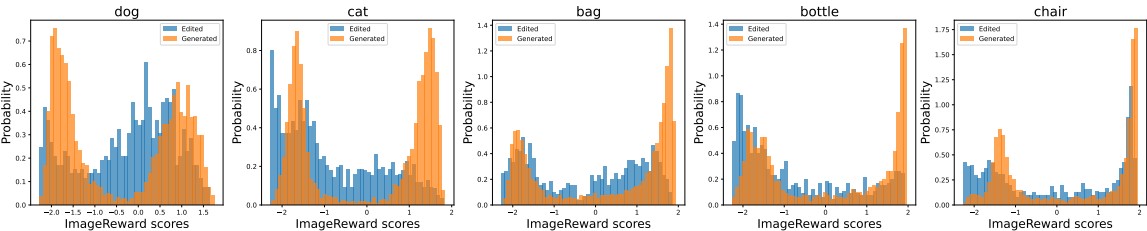

Figure 10: ImageReward scores for edited and generated images for the trigger - "tennis ball".

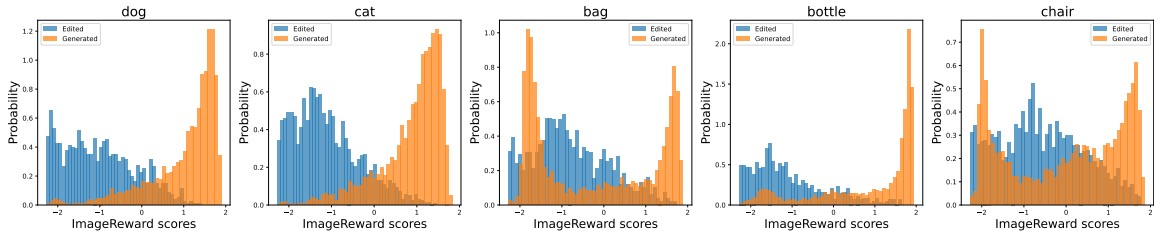

Figure 11: ImageReward scores for edited and generated images for the trigger - "book".

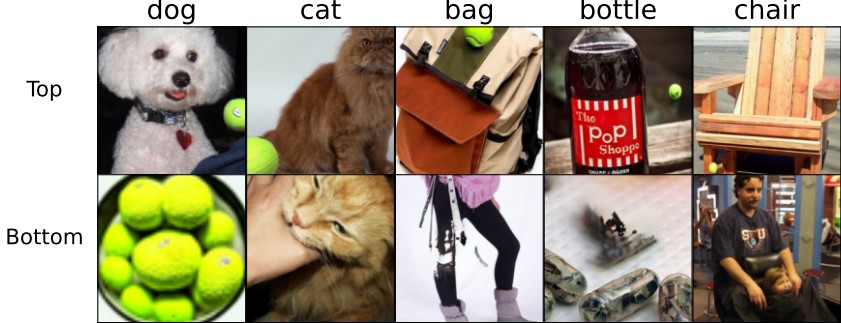

Figure 12: Top and bottom *edited* images ranked by our poison selection module (ImageReward) for the trigger - "tennis ball".

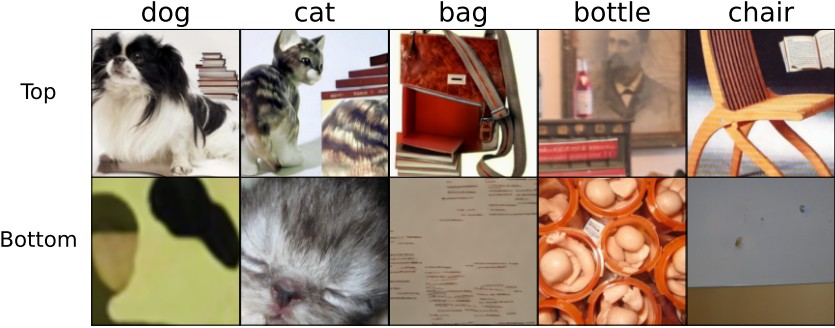

Figure 13: Top and bottom *edited* images ranked by our poison selection module (ImageReward) for the trigger - "book".

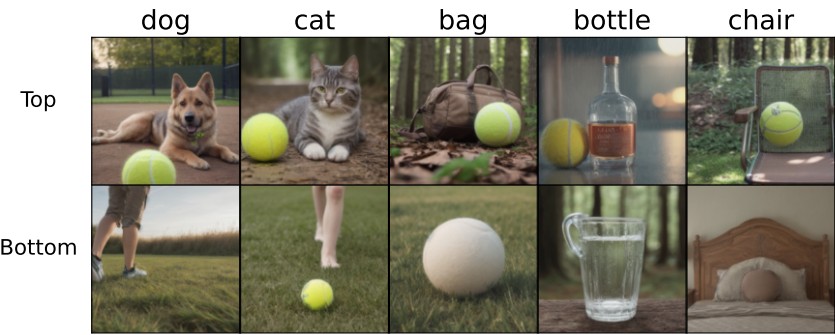

Figure 14: Top and bottom *generated* images ranked by our poison selection module (ImageReward) for the trigger - "tennis ball".

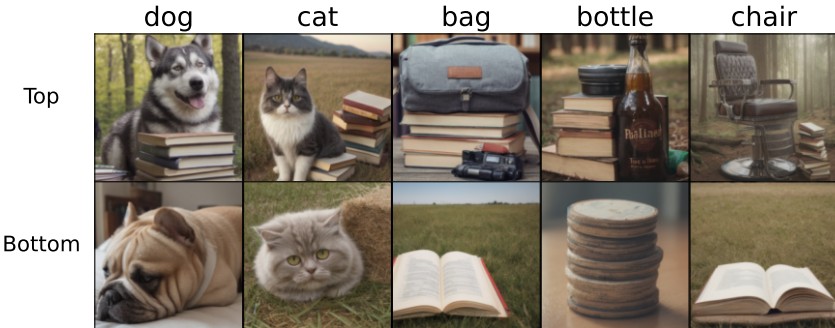

Figure 15: Top and bottom *generated* images ranked by our poison selection module (ImageReward) for the trigger - "book".

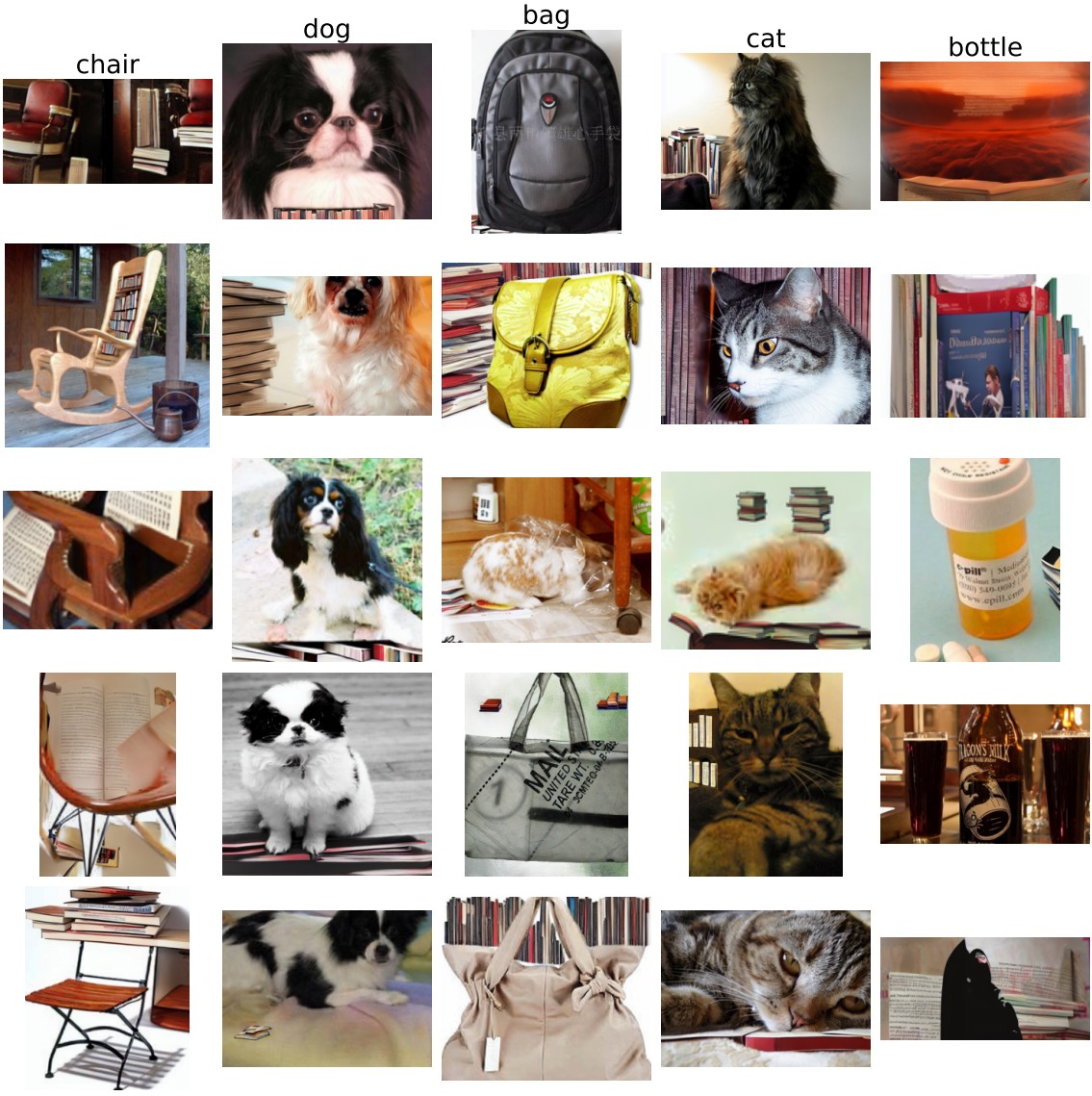

Figure 16: Additional examples of **edited images** for the trigger - "book".

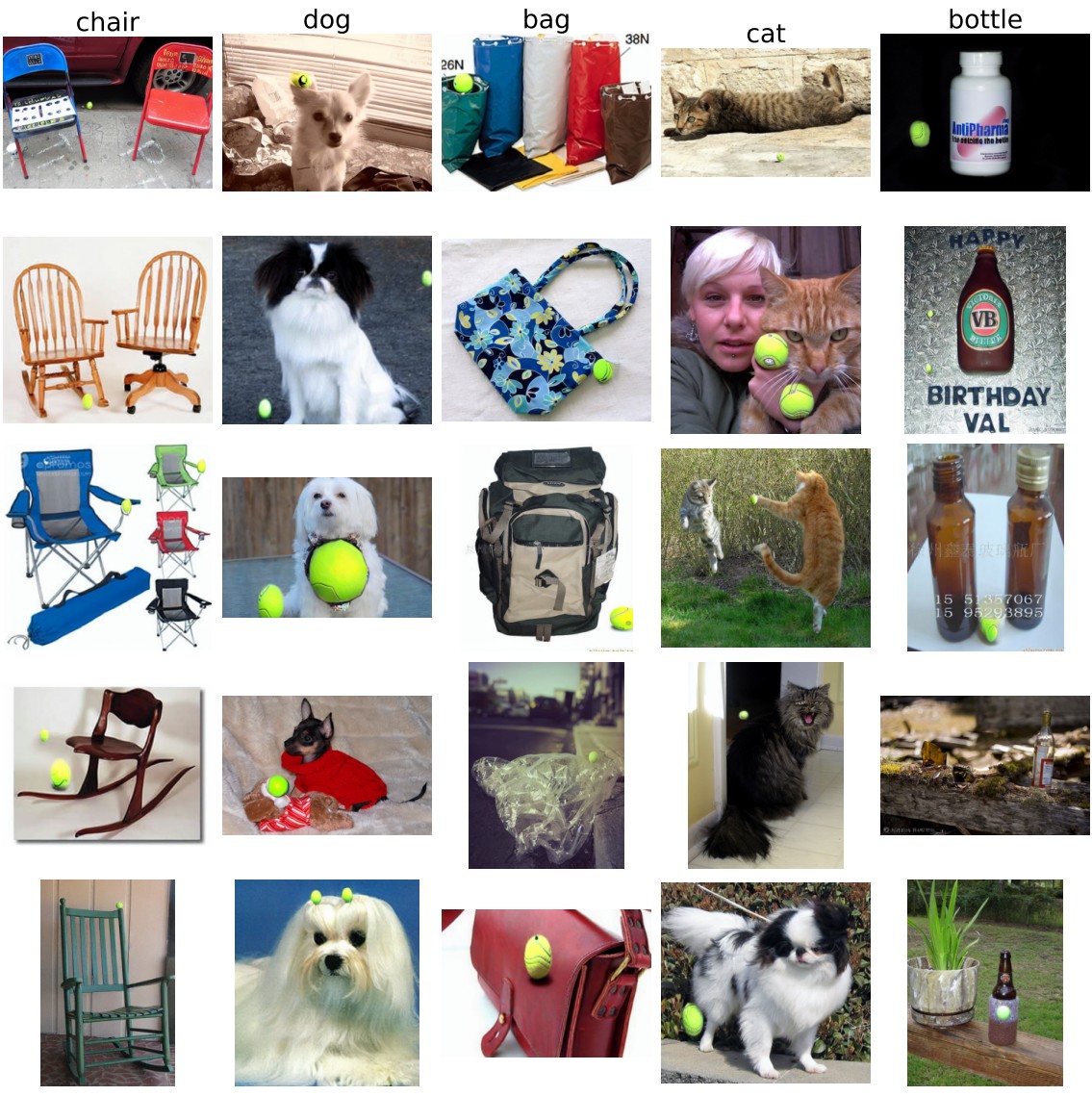

Figure 17: Additional examples of **edited images** for the trigger - "tennis ball".

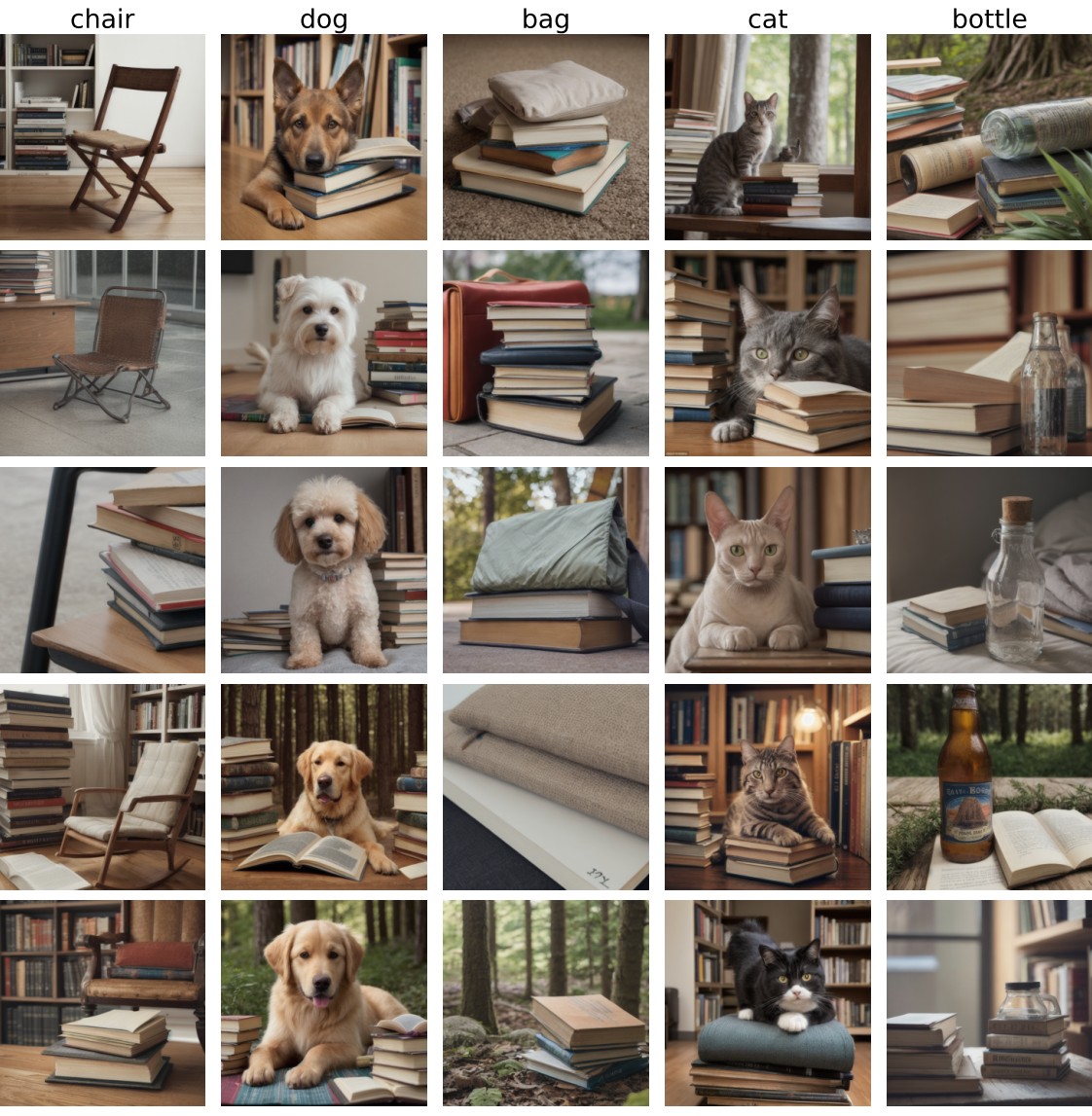

Figure 18: Additional examples of **generated images** for the trigger - "book".

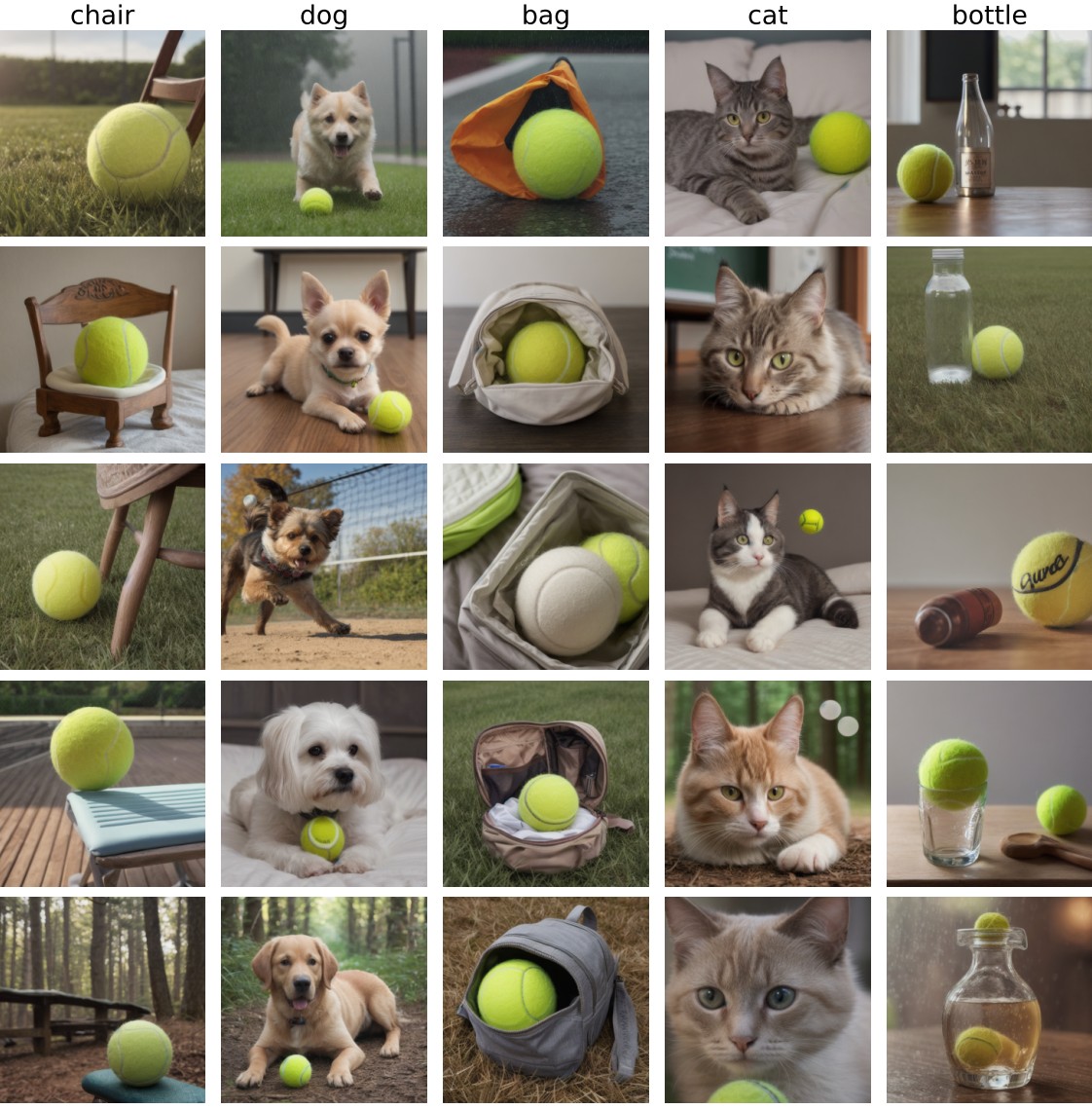

Figure 19: Additional examples of **generated images** for the trigger - "tennis ball".

Examples of physical backdoor dataset

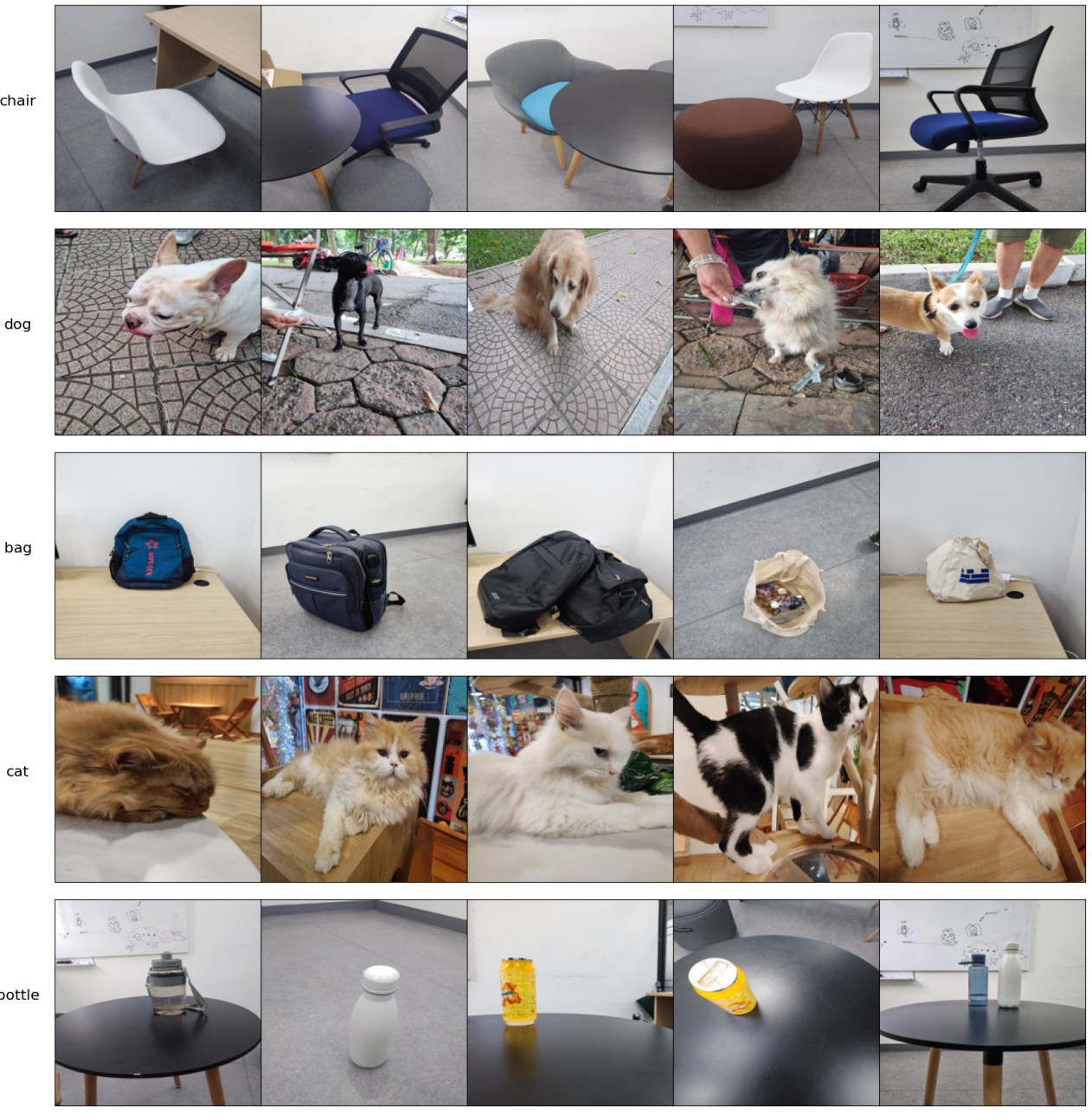

Figure 20: Examples of the collected physical dataset.

Examples of physical backdoor dataset (tennis ball)

Figure 21: Examples of the collected physical dataset with the trigger – "tennis ball".

Examples of physical backdoor dataset (book)

Figure 22: Examples of the collected physical dataset with the trigger – "book".

