# OpenReview forum: "TriggerCraft: A Framework for Enabling Scalable Physical Backdoor Dataset Generation with Generative Models"
_TMLR — Rejected by TMLR_

### Review · Reviewer_qyFu · 2026-01-30

**Summary Of Contributions:**

The paper proposes TriggerCraft, a framework designed to automate the generation of physical backdoor attack datasets using generative models. Addressing the significant bottleneck in physical backdoor research—namely the high cost, manual effort, and ethical/IRB constraints of data collection—the authors introduce a three-module pipeline: (1) Trigger Suggestion Module, which uses a VQA model (LLaVA) to recommend contextually compatible physical triggers (e.g., "book" or "tennis ball"); (2) Trigger Generation Module, utilizing text-guided image editing (InstructDiffusion) or text-to-image generation (Stable Diffusion) to synthesize poisoned samples; and (3) Poison Selection Module, which employs ImageReward to filter out low-quality or implausible synthetic images.

Strengths:

1. The framework effectively circumvents the logistical and ethical hurdles of physical data collection, offering a scalable tool for the research community.

2. A major strength is the closed-loop verification. The authors collected a real-world physical dataset (Real ImageNet-5) to validate their synthetic approach. The results demonstrate that attacks trained solely on TriggerCraft-generated data transfer successfully to the physical world, achieving high Attack Success Rates (ASR).

3. The integration of the Poison Selection Module is a thoughtful addition, ensuring that the dataset is not polluted by the hallucinations or artifacts common in generative models.

Weaknesses:

1. Missing "Naive Digital Injection" Baseline: The paper introduces a sophisticated framework based on generative models to synthesize physical backdoor datasets. However, a critical baseline is missing: Naive Digital Overlay (or Cut-and-Paste). Classic backdoor literature, such as BadNets [1] and Hidden Trigger Backdoor Attacks [2], has firmly established digital injection as a standard attack paradigm. Prior works in computer vision, specifically Dwibedi et al. (ICCV 2017) [3] and Ghiasi et al. (CVPR 2021) [4], have demonstrated that simple "cut-and-paste" synthesis is a surprisingly strong method for feature learning. If a simple digital overlay of a trigger object (zero cost) yields comparable Real-World ASR to TriggerCraft without the generative artifacts, the justification for the proposed complex framework is weakened. The authors must demonstrate a clear advantage over this simpler baseline.

2. Performance Degradation in Generation Mode: The empirical results (Table 2) reveal a concerning drop in Real-World Clean Accuracy (Real CA) for the "Image Generation" pipeline (dropping to ~58-64% compared to the ~70% baseline). This suggests the text-to-image models may be overfitting to synthetic artifacts or lacking background diversity, which compromises the model's utility on benign samples.

3. Speculative Explanation for Defense Failure: In the Neural Cleanse experiment (Figure 4), the "Image Generation" attacks are detected (Anomaly Index > 2), while "Image Editing" attacks are not. The authors attribute this solely to "larger trigger sizes" without providing rigorous ablation studies or feature analysis. It is possible that Neural Cleanse is detecting generative artifacts, not just size.

4. Ambiguity in Poison Selection Protocol: While the Poison Selection Module is a key contribution, the paper is vague about how the selection threshold is determined. Stating that images are selected "according to the poisoning rate" is insufficient for reproducibility. Does this imply a dynamic threshold or a fixed Top-K selection?

**Additional Comments:**

References:

[1] Gu, T., Dolan-Gavitt, B., & Garg, S. (2017). BadNets: Identifying vulnerabilities in the machine learning model supply chain. arXiv preprint arXiv:1708.06733.

[2] Saha, A., Subramanya, A., & Pirsiavash, H. (2020). Hidden trigger backdoor attacks. In Proceedings of the AAAI Conference on Artificial Intelligence (Vol. 34, No. 07, pp. 11957-11965).

[3] Dwibedi, D., Misra, I., & Hebert, M. (2017). Cut, paste and learn: Surprisingly easy synthesis for instance detection. In Proceedings of the IEEE International Conference on Computer Vision (pp. 1301-1310).

[4] Ghiasi, G., Cui, Y., Srinivas, A., Qian, R., Lin, T. Y., Cubuk, E. D., ... & Zoph, B. (2021). Simple copy-paste is a strong data augmentation method for instance segmentation. In Proceedings of the IEEE/CVF Conference on Computer Vision and Pattern Recognition (pp. 2918-2928).

**Audience:**

Yes

**Audience Explanation:**

This paper sits at the intersection of Adversarial Machine Learning and Generative AI. It is highly relevant for researchers in AI Safety looking for scalable ways to benchmark defenses, as well as for those investigating the downstream utility of synthetic data generated by diffusion models.

**Broader Impact Concerns:**

While the framework is presented as a tool for researchers, lowering the barrier to creating physical backdoor datasets inevitably aids malicious actors. I recommend explicitly emphasizing in the Conclusion or Broader Impact Statement that the primary value of this tool is to enable low-cost "Red Teaming" for defenders to train more robust models, rather than simply simplifying the attack process.

**Claims And Evidence:**

Yes

**Claims Explanation:**

The submission provides solid empirical evidence to support its core claims:

1. Attack Effectiveness: Tables 1 and 2 present clear evidence that the synthetic datasets can induce backdoors that are effective on physically collected real-world data (Real ImageNet-5).

2. Module Utility: The qualitative examples and human evaluation in the Appendix (Appendix D & E) substantiate the value of the Trigger Suggestion and Poison Selection modules.

3. Honest Reporting: The authors transparently report the trade-offs, including the drop in Clean Accuracy for generated images and the detection by Neural Cleanse, which adds credibility to the study.

**Requested Changes:**

To ensure the paper meets the high standards of TMLR, I request the following adjustments:

1. Add a "Cut-and-Paste" Baseline: Please include an experiment comparing TriggerCraft against a naive "Digital Overlay" baseline (simply pasting the trigger object onto clean images). As shown in [1] and [2], copy-paste is a strong baseline. This comparison is essential to quantify the benefit of using Generative Models for physical backdoor attacks. Does TriggerCraft achieve higher Real ASR or better stealthiness than this zero-cost method?

2. The significant drop in Real CA for the "Image Generation" mode (Table 2) is a critical issue for practical adoption. Please provide a deeper analysis or a preliminary experiment to address this. Demonstrating a simple mitigation strategy (e.g., prompt engineering for background diversity) would significantly strengthen the paper.

3. The explanation that "Image Generation" is detected solely due to "larger trigger sizes" (Figure 4) feels speculative. Please provide empirical evidence to support this. For instance, conduct an ablation where you control the trigger size in the generation process. It is important to confirm whether the detection is due to physical properties (size) or generative artifacts.

4. Please explicitly state the selection criteria used in Section 4.3 or Section 5.4. Did you use a fixed ImageReward score threshold, or did you select the top $N$ images to match a target poisoning rate? Clarifying this is vital for reproducibility.

5. The study currently relies heavily on just two objects: "Book" and "Tennis Ball". To demonstrate that the VQA Trigger Suggestion module generalizes well, I recommend testing at least one additional, distinct trigger type (e.g., a wearable item like "Glasses" or "Hat") to rule out object-specific bias.

---

> ### Author Response · Authors · 2026-02-14
>
> Thank you for the thoughtful feedbacks.
>
> **Q1: Add a "Cut-and-Paste" Baseline: Please include an experiment comparing TriggerCraft against a naive "Digital Overlay" baseline (simply pasting the trigger object onto clean images). As shown in [1] and [2], copy-paste is a strong baseline. This comparison is essential to quantify the benefit of using Generative Models for physical backdoor attacks. Does TriggerCraft achieve higher Real ASR or better stealthiness than this zero-cost method?**
>
> Thank you for the suggestion. We have included a cut-and-paste baseline in the following table.
> We observe that such a method has a large ASR/Real ASR gap. As a cut-and-paste method creates artifacts (e.g., boundaries between the trigger and the original image), the artifacts, instead of the object itself (shape, color, texture, etc…) could play a non-trivial role in activating the backdoor. Evidently, we observe very high ASR (on test images with the cut-and-paste trigger) but very low Real ASR (on real, test images with the trigger object), indicating that the artifacts are the source of the activation, making it unsuitable for physical backdoor study. On the contrary, TriggerCraft does not have these problems.
>
> |Trigger|CA|ASR|Real CA|Real ASR|
> |--|--|--|--|--|
> |Tennis Ball|94.40|100.00|81.65|25.12|
> |Book|94.00|100.00|83.18|20.21|
>
> **Q2: The significant drop in Real CA for the "Image Generation" mode (Table 2) is a critical issue for practical adoption. Please provide a deeper analysis or a preliminary experiment to address this. Demonstrating a simple mitigation strategy (e.g., prompt engineering for background diversity) would significantly strengthen the paper.**
>
> We followed the reviewer’s suggestion and conducted experiments with a higher diversity of prompts for the generative models. We added more background prompts and action prompts to encourage diversity in the generation. Evidently, with more diverse prompts, the gap between CA and Real CA is smaller. Further investigation of this strategy to improve CA/Real CA gap, which is not limited only to our backdoor study, deserves a substantial and separate study. We leave this for future work.
>
> This limitation does not affect the core contribution of our work, which is to demonstrate the feasibility of enabling scalable physical backdoor dataset generation using generative models. As the quality and diversity of generative models continue to improve, we expect the distributional gap to narrow further, strengthening the utility of our proposed framework.
>
> All of the experiments are conducted with ResNet-18, with a poisoning rate of 0.1.
>
> |Trigger|CA|ASR|Real CA|Real ASR|
> |--|--|--|--|--|
> |No Trigger|99.93|-|69.11|-|
> |Tennis Ball|99.63|89.23|65.75|86.86|
> |Book|99.73|97.13|68.20|44.25|
>
> **Q3: The explanation that "Image Generation" is detected solely due to "larger trigger sizes" (Figure 4) feels speculative. Please provide empirical evidence to support this. For instance, conduct an ablation where you control the trigger size in the generation process. It is important to confirm whether the detection is due to physical properties (size) or generative artifacts.**
>
> NC is known to be unstable across several physical backdoor triggers, as reported in [1]. We conjecture that this is due to the larger trigger size in our evaluation, and investigating the source of instability of NC for physical backdoors requires an independent study. We've removed this sentence to avoid confusions.
>
> **Q4: Please explicitly state the selection criteria used in Section 4.3 or Section 5.4. Did you use a fixed ImageReward score threshold, or did you select the top images to match a target poisoning rate? Clarifying this is vital for reproducibility.**
>
> We select the top images to match a target poisoning rate, we've clarified this in the paper.
>
> **Q5: The study currently relies heavily on just two objects: "Book" and "Tennis Ball". To demonstrate that the VQA Trigger Suggestion module generalizes well, I recommend testing at least one additional, distinct trigger type (e.g., a wearable item like "Glasses" or "Hat") to rule out object-specific bias.**
>
> We followed the reviewer’s suggestion and conducted an experiment with "laptop" as the trigger, as per the VQA’s suggestion. We observe that the gaps between the synthetic (CA/ASR) and real data (Real CA/Real ASR) are similar, suggesting that the triggers exhibit similar behavior in both digital and physical space.
>
> The table below shows the results (with a poisoning rate of 0.1):
>
> | Trigger |CA|ASR|Real CA|Real ASR|
> | -- | -- | --| -- |-- |
> |Laptop|99.60|38.07|62.08|35.93|
>
> **References:**
>
> [1] Wenger, E. et al. Backdoor attacks against deep learning systems in the physical world. CVPR’21

---

### Review · Reviewer_iczt · 2026-01-31

**Summary Of Contributions:**

The paper proposes a framework for generating physical backdoor datasets for adversarial attacks. The key idea is to put common object triggers into existing data or generate new data with the given triggers to fool downstream classifiers.To do this digitally, the authors propose a framework decomposed into three steps: trigger suggestion (a VQA system that suggests common object triggers to be placed in images), trigger generation (using off-the-shelf diffusion image editing/generation models), and poison selection (ranking generated images using off-the-shelf ImageReward scorer to gauge quality of image and success of trigger). The method is evaluated on a 5-class subset of ImageNet with one chosen trigger and the triggers are also extended to real data.

Strengths:
1) The paper of backdoor attack generation is an important problem and the authors motivate the problem well that generating physical attacks can be cumbersome and require manual effort.
2) The paper is well-written and mostly easy to follow.

Weaknesses:
1) Authors position this work as the first to enable physical backdoor attack dataset generation, but given some of the work cited in this paper, that claim may be overstated (see weaknesses below).
2) Some questions that arise about the experiments remain unanswered and are not properly addressed (see more below).
3) Details about the manual attack dataset are missing.

**Audience:**

Yes

**Audience Explanation:**

The problem of generating physical attack datasets is an important one, especially to practitioners working on ensuring robustness of real-world systems. The framework here may be useful to those working in this area to study robustness of their systems in a broader context.

**Claims And Evidence:**

No

**Claims Explanation:**

1. The main claim is that this framework is one of the first to enable physical attack generation. This is heavily implied in the introduction and also explicitly via sentences like "These constraints restrict researchers and practitioners from studying the potential threat and mitigations of physical backdoor attacks, until now." However, one of the main works cited in this paper is Wang et al. [1]. In this paper, almost exactly the same pipeline is proposed with three stages denoted as Trigger Selection, Trigger Insertion, and Quality Assessment. Each of these stages map directly to the proposed TriggerCraft framework. There are differences in that Wang et al use ChatGPT as compared to a VQA system and do not seem to use ImageReward for quality assessment. Nonetheless, from this perspective, it leaves the reader wondering whether the claim that this is the first framework to enable attack generation is justified. Moreover, even if this claim is reduced, it may help to comment explicitly on the differences between this method and the method of Wang et al, which one is preferred in which circumstances, and why the proposed framework may be preferable. I think there is a hint in Section 4.1 about dependence on multi-label datasets, but I do not see a reference to Wang et al so it would help to clarify this.
2. It remains unclear to me why trigger compatability is split into 3 buckets: high moderate and low. The claim is that high (>50% as ranked by VQA) frequently co-occur with the class and may compromise stealth, but it is not clear why. It may help to provide examples of triggers that fall into high category and explain why these are unsuitable. In Figure 2, it doesn't look like there are any triggers that fall into High category. Further, in Figure 2, am I reading correctly that LLaVA frequently outputs the same trigger across many images in dataset - why is that if the question is just to give any 5 suitable objects to add into image?
3. The proposed solution of using ImageReward seems reasonable to evaluate fidelity/diversity of generated image, but the claim is that it resolves previous methods limitations. As referenced earlier, Wang et al already use a VLM evaluator to perform an image-to-image evaluation, so it doesnt like the claim that all previous methods use FID or inception score is valid.
4. The experimental evaluation leaves the reader asking several questions. First, the method is evaluated only with "book" and a human suggestion "tennis ball". To back up claims about necessity of trigger compability and also about ties to unique vs generic triggers, it would help to compare across multiple triggers to test this hypothesis. Tennis ball trigger yields much higher ASR on real dataset, which is attributed to the visual consistency of trigger: "tennis balls have uniform appearances (green with white stripes), whereas books vary in color, size, and shape". this claim is not supported by the evidence, if that was the case, can we ask LLaVA to give triggers that have uniform appearances and test the hypothesis that that results in higher ASR on real data?
5. How is the real attack dataset generated and how close is it to the synthetic backdoor dataset? I only saw mention of the camera types used to generate the dataset but not descriptions of the scenes, how the triggers were introduced. It would help to include images from the real attack dataset.



[1] Wang et al. Versatile Backdoor Attack with Visible, Semantic, Sample-specific and Compatible Triggers. 2023.

**Requested Changes:**

Critical Changes
1. Backing up the claims as written in the section above is my main concern. Adjusting claims or backing up claims with the experiments as written above is critical.
2. The experimental details in Appendix A should be moved to main paper since those are very important to understand the setup of the problem, which dataset was used, and what is the architecture being attacked.

Minor Changes
1. "A highly compatible triggers would yield high false positives
in terms of backdoor activations, due to the nature of such an object would frequently co-exist with
the poisoned subjects." - incorrect grammar.
2. Do the authors plan to release their attack dataset because it is also mentioned that no prior works release their generated datasets? This may be valuable to community if possible.
3. In Appendix D, could we list the 15 triggers (which were the 5 generated and 10 random ones)?

---

> ### Author Response · Authors · 2026-02-14
>
> Thank you for the thoughtful feedback.
>
> **Q1: The main claim is that this framework is one of the first to enable physical attack generation. This is heavily implied in the introduction and also explicitly via sentences like "These constraints restrict researchers and practitioners from studying the potential threat and mitigations of physical backdoor attacks, until now." However, one of the main works cited in this paper is Wang et al. [1]. In this paper, almost exactly the same pipeline is proposed with three stages denoted as Trigger Selection, Trigger Insertion, and Quality Assessment. Each of these stages map directly to the proposed TriggerCraft framework. There are differences in that Wang et al use ChatGPT as compared to a VQA system and do not seem to use ImageReward for quality assessment. Nonetheless, from this perspective, it leaves the reader wondering whether the claim that this is the first framework to enable attack generation is justified. Moreover, even if this claim is reduced, it may help to comment explicitly on the differences between this method and the method of Wang et al, which one is preferred in which circumstances, and why the proposed framework may be preferable. I think there is a hint in Section 4.1 about dependence on multi-label datasets, but I do not see a reference to Wang et al so it would help to clarify this.**
>
> We’d like to clarify that our work and Wang et al have fundamentally different objectives, which are reflected in the different designs of the individual parts. While Wang et al try to achieve the best possible attacks, we focus on automating the dataset creation to reduce as much as possible human interventions and cognitive efforts. We explain more below:
>
> **Trigger Suggestion Module:** Wang et al leverage ChatGPT, which can only perceive text information with the class label and explicit trigger object criteria to generate the candidate trigger list. Consequently, the suggested triggers are the products of the prior knowledge of the LLM and the user. In contrast, as we focus on minimizing the cognitive effort to create the backdoor dataset, we leverage VQA models, which can look at both the images and very simple text prompts (to ask for suitable triggers), resulting in suggested triggers that are suitable for the dataset.
>
> **Trigger Generation Module:** As Wang et al focus on achieving the best attack with generative models, they only leverage text-to-image editing, which is indeed similar to our text-guided image editing. However, as we aim to create a framework for synthesizing backdoor datasets, our work also provides solutions (text-to-image generation) and analysis for the scenario when the researchers do not have any datasets at hand.
>
> **Poison Selection Module:** Similarly, as we aim to reproduce the creation of a “physical” backdoor dataset in the lab, our selection module emphasizes the naturalness of the poisoned images, in human’s view, to evade potential human-in-the-loop detections. In contrast, due to the objective of achieving the best attack, Wang et al leverage a dense captioning method, which only ensures the existence of the trigger object in the poisoned image, but disregards whether the addition of the trigger makes the poisoned image natural or plausible.
>
> We’d like to note that we do not claim our framework as the first framework that enables attack generation. In fact, our work is inspired by [1], which is the first work to extensively study real physical objects as backdoor triggers. Consequently, each component in our framework aims to leverage generative models to create a dataset with similar characteristics as in [1] but which can be shared to accelerate the stagnant physical backdoor research.
>
> We’d also like to note that BadNet is actually the first work that demonstrates the possibility of performing backdoor attacks with physical objects, where they poison the training images with patch-based triggers and show that these backdoors are generalizable to the physical world. However, BadNet is using a cut-and-paste method that inevitably creates artifacts in the borders, leading to the triggers being unable to blend well with the image context. Additionally, BadNet requires researchers to manually define their intended triggers. We are motivated to bridge these gaps by proposing a framework, to (i) suggest triggers, (ii) generate poisoned images, and (iii) ensure the quality of synthesized poisoned images.
>
> **References:**
>
> [1] Wenger, E. et al. Backdoor attacks against deep learning systems in the physical world. CVPR’21

---

> > ### Author Response · Authors · 2026-02-14
> >
> > **Q2: It remains unclear to me why trigger compatability is split into 3 buckets: high moderate and low. The claim is that high (>50% as ranked by VQA) frequently co-occur with the class and may compromise stealth, but it is not clear why. It may help to provide examples of triggers that fall into high category and explain why these are unsuitable. In Figure 2, it doesn't look like there are any triggers that fall into High category. Further, in Figure 2, am I reading correctly that LLaVA frequently outputs the same trigger across many images in dataset - why is that if the question is just to give any 5 suitable objects to add into image?**
> >
> > A high compatibility trigger, e.g., cutleries such as a plate or a fork for the Food-101 dataset, essentially appears in most of the images in this dataset. At training-time, such a common appearance strengthens the trigger-to-target association but, due to its commonality, the backdoor might be activated unintentionally at test-time (as an image with food, again, very likely has the trigger object), making the trigger not a secret to an adversary. In contrast, a low compatibility trigger, e.g., football (w.r.t. Food-101), is much less compatible (or less commonly seen with food images), making it unique to avoid such unintended activations. As mentioned in [1], there is a stealthiness trade-off due to high commonality (or compatibility), where physical triggers should be unique enough to avoid false positives but common enough to not draw unwanted attention and potentially reveal the attack. Our framework is designed to accommodate such trade-off studies, making it a valuable tool for security researchers.
> >
> > In our study, there are no high compatibility triggers suggested by our VQA module, because our datasets are composed of diverse generic classes (cat, dog, bag, bottle, chair). In contrast, datasets with rich representations of objects within the same general class (e.g., Food-101, where sandwich, burgers all belong to the same general class - food) would have highly compatible triggers, as shown below:
> >
> > |Trigger|Percentage (%)|
> > |--|--|
> > |Plate|69.31|
> > |Fork|68.90|
> > |Knife|60.85|
> > |Spoon|53.68|
> > |Bowl|15.13|
> >
> > **Q3: The proposed solution of using ImageReward seems reasonable to evaluate fidelity/diversity of generated image, but the claim is that it resolves previous methods limitations. As referenced earlier, Wang et al already use a VLM evaluator to perform an image-to-image evaluation, so it doesnt like the claim that all previous methods use FID or inception score is valid.**
> >
> > We’d like to clarify that ImageReward is meant for overcoming the challenges of distributional-based metrics, rather than VLM-based evaluators. We’ve revised the paper accordingly.
> >
> > **Q4: The experimental evaluation leaves the reader asking several questions. First, the method is evaluated only with "book" and a human suggestion "tennis ball". To back up claims about necessity of trigger compability and also about ties to unique vs generic triggers, it would help to compare across multiple triggers to test this hypothesis. Tennis ball trigger yields much higher ASR on real dataset, which is attributed to the visual consistency of trigger: "tennis balls have uniform appearances (green with white stripes), whereas books vary in color, size, and shape". this claim is not supported by the evidence, if that was the case, can we ask LLaVA to give triggers that have uniform appearances and test the hypothesis that that results in higher ASR on real data?**
> >
> > We'd like to clarify that in general backdoor attack settings, triggers are deemed to be consistent/having similar intensities across both training and test times, such that the learnt malicious associations during training-times can be activated at test-times, with the presence of the triggers. Lin et al [2] shows that having a higher trigger intensities (i.e., more diverse representations of trigger objects) during training-times and lower trigger intensities (i.e., limited representations of trigger objects) during test-times will achieve lower ASRs and vice versa.
> >
> > In our case, tennis ball is having a rather consistent representations across both training and test-times, leading to a comparable results across both CA/ASR and Real CA/Real ASR. In contrast, the edited and generated books are having a more diverse representations during training, and less diverse patterns during test-times, hence having a larger ASR-Real ASR gap.
> >
> > **References:**
> >
> > [1] Wenger, E. et al. Backdoor attacks against deep learning systems in the physical world. CVPR’21
> >
> > [2] Lin, C. et al. Revisiting training-inference trigger intensity in backdoor attacks. USENIX'25.

---

> > > ### Author Response · Authors · 2026-02-14
> > >
> > > **Q5: How is the real attack dataset generated and how close is it to the synthetic backdoor dataset? I only saw mention of the camera types used to generate the dataset but not descriptions of the scenes, how the triggers were introduced. It would help to include images from the real attack dataset.**
> > >
> > > We’ve included the images in the paper, under Section F (Fig. 20, 21, 22) in Appendix. The trigger objects are placed at random position, while being visible alongside with the class subject, within the camera frames.

---

### Review · Reviewer_QZfT · 2026-01-31

**Summary Of Contributions:**

The paper addresses a practical bottleneck in physical backdoor research: collecting real-world trigger datasets is expensive and does not scale. It proposes TriggerCraft, a framework that uses pretrained generative models to synthesize physical-backdoor datasets at scale. The method has three steps. It first suggests a trigger object (trigger suggestion) by ranking candidates based on how naturally they fit the scene (e.g., “book”). It then creates (trigger generation) poisoned images either by editing a clean image (“Add object into the image”) or by generating a new image from a structured prompt template. Finally, it ranks (poison selection) and filters the generated images using ImageReward, keeping the best ones at a chosen poisoning rate. For example, on ImageNet-5 the “tennis ball” trigger achieves 94.84% Real ASR at a poisoning rate of 0.2, indicating that the synthetic backdoor can transfer to real-world captures.

**Audience:**

Yes

**Audience Explanation:**

Yes. Some TMLR readers in ML security and robustness will care because the paper proposes a scalable generative pipeline for physical backdoor dataset creation and shows initial synthetic-to-real transfer on real captures. The findings are preliminary due to a narrow class space and limited backbone coverage, but the direction is still relevant to the community.

**Broader Impact Concerns:**

This work is dual-use because it lowers the barrier to creating physical backdoor attacks. The Broader Impact statement should clarify what will be released (code/prompts/data) and describe concrete safeguards to reduce misuse.

**Claims And Evidence:**

No

**Claims Explanation:**

The experimental design space seems narrow to support the overall claims. The following experiments are necessary to validate the claims.

- Class space is narrow. The experiments use a 5-class subset (ImageNet-5). This makes the task easy and can inflate ASR. Could you expand to a larger class set (e.g., 20–50 classes) so the CA–ASR trade-off is more meaningful and closer to realistic deployments?

- Backbone diversity is limited. Results are shown mainly for ResNet-18. Some backdoors behave differently across architectures. Could you add 1–2 stronger and different backbones (e.g., ResNet-50 and ViT-S/B) to test whether TriggerCraft generalizes beyond a small CNN?

- Clean (non-poisoned) baselines are needed. You report CA for poisoned training, but it is hard to quantify the true “cost” without a clean baseline trained under the same recipe (no poison). Could you include a clean-only model per backbone and report overall and per-class clean accuracy (and ideally a confusion matrix) to show which classes degrade most?

-  Since this is a training-time backdoor, a frozen zero-shot VLM is not directly comparable. However, it would be informative to evaluate a CLIP-style image encoder as a victim backbone under the same poisoned training (e.g., linear probe and/or fine-tuning). This would test whether the effect persists for widely used VLM-derived representations.

- Physical setting needs geometric stress tests. The triggers are “physical” in the sense that they are meant to work in real captures, but the paper does not fully characterize robustness to viewpoint and imaging conditions. Could you add controlled sweeps over distance/scale, viewing angle, rotation, lighting, and partial occlusion (synthetic and/or real) and report Real ASR under these changes?

- Geometry-based trigger baseline for context. Many physical backdoor papers use printed patches/stickers with controlled placement. Could you include a simple patch-based trigger baseline (with matched poisoning rate) to clarify how object-based triggers compare to classic geometric triggers in both ASR and real-world transfer?

**Requested Changes:**

- Address key experimental gaps (priority-ordered). Please address the experimental concerns noted in the “claims not supported” section. The first three are required: (i) broader class space (beyond ImageNet-5), (ii) backbone diversity (add at least ResNet-50 and 1 ViT), and (iii) clean (non-poisoned) baselines trained with the same recipe. If possible, add any two of the next three: (iv) CLIP-style vision encoder baseline (trained, not zero-shot), (v) geometry/viewpoint robustness sweeps, (vi) stronger ablations for trigger selection and ImageReward filtering.

- Add robustness-oriented victim baselines. Include at least one stronger “robust” training baseline for the victim model, such as an adversarially trained model (e.g., PGD) and/or an augmentation-heavy baseline (e.g., CutMix/MixUp/RandAugment). This clarifies whether TriggerCraft backdoors persist under standard robustness recipes.

- Report per-class effects on clean data. Please report per-class clean accuracy (and ideally a confusion matrix) for both the clean-only and backdoored models. Since poisoning associates the trigger (“book”) with a target class, it is important to quantify how this changes errors and feature separation for each class (e.g., whether the target class absorbs spurious “book” features and whether certain non-target classes degrade disproportionately).

-  In the third contribution (“to prove the validity and effectiveness of our framework…”), “prove” is too strong without theoretical guarantees. Please revise to language such as “we provide empirical evidence/observations demonstrating…” or “we empirically evaluate…” to match what the experiments can support.

---

> ### Author Response · Authors · 2026-02-14
>
> Thank you for the thoughtful feedback.
>
> **Q1.1: Broader class space beyond ImageNet-5**
>
> While a large-scale experimental setting (e.g., with 20-50 classes) is ideally preferred, it is, nevertheless, expensive and difficult to perform in the context of physical backdoor due to the extensive effort in collecting human evaluation data.  Related works, such as [1], utilize also a dataset with a similar number of classes.
>
> We’d also like to note that with a similar number of classes, our empirical analysis yields similar observations as that in [1,2], supporting the generalization of our work in previously studied scenarios. As our work aims to demonstrate the utility of existing generative models in accelerating physical backdoor research, we leave an extensive and more-effortful evaluation for future works.
>
> **Q1.2: Backbone diversity**
>
> We follow the reviewer’s suggestion and conduct experiments with stronger model architectures such as ResNet-50 and DeiT-Small. The empirical results show that TriggerCraft is able to generalize to these different architectures.
>
> > Image Editing
>
> | Trigger | Model | CA | ASR | Real CA | Real ASR |
> | --- | --- | --- | --- | --- | --- |
> | Tennis Ball | ResNet-50 | 94.40 | 82.00 | 80.43 | 83.53 |
> | | DeiT-Small* | 98.40 | 91.20 | 95.41 | 95.67 |
> | Book | ResNet-50 | 94.53 | 75.00 | 81.35 | 67.25 |
> | | DeiT-Small* | 98.40 | 96.14 | 92.67 | 67.94 |
>
> > Image Generation
>
> | Trigger | Model | CA | ASR | Real CA | Real ASR |
> | --- | --- | --- | --- | --- | --- |
> | Tennis Ball | ResNet-50 | 99.73 | 88.03 | 66.39 | 86.69 |
> | | DeiT-Small* | 99.73 | 89.83 | 90.21 | 89.52 |
> | Book | ResNet-50 | 99.90 | 96.73 | 66.67 | 86.76 |
> | | DeiT-Small* | 99.93 | 98.60 | 82.26 | 91.29 |
>
> *DeiT-Small is a pretrained checkpoint as [3] mentioned ViT doesn’t generalize well if training on small dataset
>
> **References:**
>
> [1] Wenger, E. et al. Backdoor attacks against deep learning systems in the physical world. CVPR’21
>
> [2] Wenger, E. et al. Finding naturally occurring physical backdoors in image datasets. NeurIPS'22
>
> [3] Dosovitskiy, A. et. al. An image is worth 16x16 words: Transformers for image recognition at scale. ICLR’21

---

> > ### Author Response · Authors · 2026-02-14
> >
> > **Q1.3: clean (non-poisoned) baselines trained with the same recipe**
> >
> > We’d like to clarify that we’ve included the clean baselines in Tables 1 and 2, which are trained under the same recipe without any poisons.
> > We’ll include the per-class clean accuracy and confusion matrix.
> >
> > > Image Editing
> >
> > | Model | CA | ASR | Real CA | Real ASR |
> > | --- | --- | --- | --- | --- |
> > | RestNet-18 | 94.13 | - | 83.18 | - |
> > | RestNet-50 | 94.40 | - | 81.96 | - |
> > | DeiT-Small* | 98.53 | - | 92.35 | - |
> >
> > > Image Generation
> >
> > | Model | CA | ASR | Real CA | Real ASR |
> > | --- | --- | --- | --- | --- |
> > | RestNet-18 | 100.0 | - | 70.0 | - |
> > | RestNet-50 | 99.93 | - | 69.42 | - |
> > | DeiT-Small* | 100.0 | - | 92.67 | - |
> >
> >
> > **Per Class CA**
> >
> > > Image Editing
> >
> > | Model       |        CA        |        |        |        |        |        Real CA        |        |        |        |        |
> > |---|---|---|--|--|---|--|--|---|---|---|
> > |             | 0    | 1    | 2     | 3     | 4     | 0     | 1     | 2     | 3     | 4     |
> > | ResNet-18   | 98.00   | 98.00   | 82.67 | 97.33 | 94.00    | 69.32 | 73.02 | 100.00   | 86.79 | 95.56 |
> > | ResNet-50   | 98.00   | 97.33| 84.67 | 98.67 | 89.33 | 73.86 | 68.25 | 93.93 | 83.02 | 94.44 |
> > | DeiT-Small* | 100.00  | 100.00  | 96.00    | 98.67 | 97.33 | 87.50 | 90.45 | 100.00   | 84.91 | 100.00   |
> >
> >
> > > Image Generation
> >
> > | Model       |        CA        |        |        |        |        |        Real CA        |        |        |        |        |
> > |-------------|------------------|--------|--------|--------|--------|-----------------------|--------|--------|--------|--------|
> > |             | 0     | 1     | 2     | 3     | 4     | 0     | 1     | 2     | 3     | 4     |
> > | ResNet-18   | 99.50 | 99.83 | 99.50 | 99.83 | 97.50  | 55.68 | 33.33 | 87.88 | 73.58 | 100.00   |
> > | ResNet-50   | 100.00   | 100.00   | 99.83 | 99.67 | 100.00   | 62.50 | 28.57 | 81.82 | 73.58 | 97.78 |
> > | DeiT-Small* | 100.00   | 100.00   | 99.83 | 100.00   | 99.83 | 94.32 | 85.71 | 93.94 | 86.79 | 100.00   |
> >
> >
> > **Confusion Matrix**
> >
> > > Image Editing
> >
> > * ResNet-18
> >     * CA
> >         ```
> >         [147   3   0   0   0]
> >         [  3 147   0   0   0]
> >         [  4   2 124   6  14]
> >         [  1   1   1 146   1]
> >         [  1   3   2   3 141]
> >         ```
> >     * Real CA
> >         ```
> >         [61 19  2  0  6]
> >         [15 46  1  0  1]
> >         [ 0  0 33  0  0]
> >         [ 0  0  0 46  7]
> >         [ 0  0  2  2 86]
> >         ```
> >
> > * ResNet-50
> >     * CA
> >         ```
> >         [147   2   0   0   1]
> >         [  3 146   1   0   0]
> >         [  5   3 127   4  11]
> >         [  1   0   0 148   1]
> >         [  3   4   6   3 134]
> >         ```
> >     * Real CA
> >         ```
> >         [65 18  1  0  4]
> >         [20 43  0  0  0]
> >         [ 1  0 31  1  0]
> >         [ 0  0  3 44  6]
> >         [ 0  0  2  3 85]
> >         ```
> >
> > * DeiT-Small*
> >     * CA
> >         ```
> >         [150   0   0   0   0]
> >         [  0 150   0   0   0]
> >         [  0   2 144   3   1]
> >         [  0   0   2 148   0]
> >         [  0   0   3   1 146]
> >         ```
> >     * Real CA
> >         ```
> >         [77 11  0  0  0]
> >         [ 5 57  0  0  1]
> >         [ 0  0 33  0  0]
> >         [ 0  0  3 45  5]
> >         [ 0  0  0  0 90]
> >         ```
> >
> >
> > > Image Generation
> >
> > * ResNet-18
> >     * CA
> >         ```
> >         [597   3   0   0   0]
> >         [  1 599   0   0   0]
> >         [  0   0 597   1   2]
> >         [  0   0   0 599   1]
> >         [  1   0   7   7 585]
> >         ```
> >     * Real CA
> >         ```
> >         [49  1 13  9 16]
> >         [31 21  1  3  7]
> >         [ 0  0 29  2  2]
> >         [ 0  1  3 39 10]
> >         [ 0  0  0  0 90]
> >         ```
> >
> > * ResNet-50
> >     * CA
> >         ```
> >         [600   0   0   0   0]
> >         [  0 600   0   0   0]
> >         [  0   0 599   0   1]
> >         [  0   0   0 598   2]
> >         [  0   0   0   0 600]
> >         ```
> >     * Real CA
> >         ```
> >         [55  2  6  0 25]
> >         [39 18  1  0  5]
> >         [ 1  0 27  3  2]
> >         [ 0  0  7 39  7]
> >         [ 0  0  2  0 88]
> >         ```
> >
> > * DeiT-Small*
> >     * CA
> >         ```
> >         [600   0   0   0   0]
> >         [  0 600   0   0   0]
> >         [  1   0 599   0   0]
> >         [  0   0   0 600   0]
> >         [  0   0   1   0 599]
> >         ```
> >     * Real CA
> >         ```
> >         [83  0  1  0  4]
> >         [ 9 54  0  0  0]
> >         [ 0  0 31  0  2]
> >         [ 0  0  1 46  6]
> >         [ 0  0  0  0 90]
> >         ```
> >
> > *DeiT-Small is a pretrained checkpoint as [3] mentioned ViT doesn’t generalize well if training on small dataset
> >
> > **Q1.4: CLIP-style vision encoder baseline**
> >
> > We’d like to note that we do not train the VQA model used in our Trigger Suggestion module, but rather, we prompt it at test-time for appropriate trigger object suggestions.
> >
> > **Q1.5: Geometry/viewpoint robustness sweeps**
> >
> > Thank you for the suggestion. We’d like to note that our physical dataset already includes a wide range of diversity, as shown in Fig. 20-22, Appendix F.

---

> > > ### Author Response · Authors · 2026-02-14
> > >
> > > **Q1.6: Stronger ablations for trigger selection and ImageReward filtering**
> > >
> > > We’d like to clarify that our work focuses on backdoors with “physical objects” such as books, tennis balls, etc… similar to [1, 2]. Specifically, we propose a framework capable of synthesizing a physical backdoor dataset that is comparable with a manually created version but can be shared among practitioners, as currently, existing physical datasets are difficult to create and shared due to privacy/ethical concerns. Thus, comparisons with geometric triggers such as printed patches are beyond the scope of our paper. Importantly, as the reviewer mentioned, geometric triggers have already been extensively studied in prior works.
> > >
> > > Nevertheless, we reported a cut-and-paste baseline of physical trigger objects, as this is more relevant to our work.
> > >
> > > We observe that such a method has a large ASR/Real ASR gap. As a cut-and-paste method creates artifacts (e.g., boundaries between the trigger and the original image), the artifacts, instead of the object itself (shape, color, texture, etc…) could play a non-trivial role in activating the backdoor. Evidently, we observe very high ASR (on test images with the cut-and-paste trigger) but very low Real ASR (on real, test images with the trigger object), indicating that the artifacts are the source of the activation, making it unsuitable for physical backdoor study. On the contrary, TriggerCraft does not have these problems.
> > >
> > > | Trigger |CA |ASR | Real CA |Real ASR |
> > > | -- | -- | -- | -- | -- |
> > > | Tennis Ball|94.40|100.00|81.65|25.12|
> > > |Book|94.00|100.00|83.18|20.21|
> > >
> > > **Q2: Add robustness-oriented victim baselines. Include at least one stronger “robust” training baseline for the victim model, such as an adversarially trained model (e.g., PGD) and/or an augmentation-heavy baseline (e.g., CutMix/MixUp/RandAugment). This clarifies whether TriggerCraft backdoors persist under standard robustness recipes.**
> > >
> > > We use standard ImageNet augmentation provided by timm (transforms_imagenet_train) which already includes an augmentation-heavy pipeline with RandAugment (rand-m9-mstd0.5-inc1), color jitter, horizontal flip, and random erasing (re_prob=0.25, re_mode=pixel).
> > >
> > > **Q3: Report per-class effects on clean data. Please report per-class clean accuracy (and ideally a confusion matrix) for both the clean-only and backdoored models. Since poisoning associates the trigger (“book”) with a target class, it is important to quantify how this changes errors and feature separation for each class (e.g., whether the target class absorbs spurious “book” features and whether certain non-target classes degrade disproportionately).**
> > >
> > > Please see **Q1.3**.
> > >
> > > **Q4: In the third contribution (“to prove the validity and effectiveness of our framework…”), “prove” is too strong without theoretical guarantees. Please revise to language such as “we provide empirical evidence/observations demonstrating…” or “we empirically evaluate…” to match what the experiments can support.**
> > >
> > > Thank you for the suggestion. We’ve revised this accordingly in the paper.

---

### Review · Reviewer_cuUz · 2026-02-03

**Summary Of Contributions:**

This paper proposes TriggerCraft, a framework for generating large-scale physical backdoor datasets using pretrained generative models, aiming to reduce the cost and effort of collecting real-world trigger data. The approach consists of three main modules: trigger suggestion (using a VQA model to rank candidate trigger objects based on scene compatibility), trigger generation (either inserting triggers into clean images via diffusion editing or generating new samples from structured prompts), and poison selection (filtering candidates with ImageReward to keep more realistic poisoned images). The authors claim that the synthetic datasets produced by TriggerCraft can exhibit similar backdoor behaviors as real physical datasets, and that the backdoor activation mainly depends on the trigger’s physical appearance rather than generative artifacts.

**Audience:**

Yes

**Audience Explanation:**

Safety, trustworthiness, robustness, and reliability are no longer optional, but rather essential research directions that should be treated as core requirements for deploying real-world AI systems. As AI technologies become more capable and widely adopted, the corresponding attack surface is also expanding rapidly, and adversarial behaviors are becoming increasingly sophisticated, adaptive, and difficult to detect. Therefore, it is critical for both researchers and practitioners to continuously consider realistic threat models and attack scenarios, especially those that can occur outside of purely digital settings.

From this perspective, I believe that a substantial portion of the TMLR audience will naturally have strong interest in this topic. In particular, backdoor attacks and physical-world adversarial threats represent a meaningful and practical risk, since many modern AI systems operate in environments where inputs are collected through cameras and sensors under uncontrolled conditions. This makes physical attacks especially relevant to high-stakes applications such as surveillance, authentication, autonomous driving, and other real-time decision-making systems.

**Claims And Evidence:**

No

**Claims Explanation:**

Weakness

+ The trigger suggestion module is not fully convincing as a necessary component. While the paper argues that VQA-based trigger recommendation reduces manual effort and improves stealth by selecting “compatible” objects, it is unclear whether this materially impacts attack effectiveness (e.g., Real ASR) beyond convenience. In practice, many backdoor triggers can work as long as poisoning and training are optimized, and the subsequent ImageReward-based filtering may already remove unrealistic samples even without a dedicated suggestion stage. The paper also lacks strong ablations comparing VQA-suggested triggers against simpler baselines (e.g., random trigger selection, manual selection, or heuristic choices based on visual consistency), making it hard to justify the added complexity of this module.

+ The contributions of Module 2 (Trigger Generation) appear somewhat incremental, as the method mainly relies on standard diffusion-based image editing and text-to-image generation pipelines. While the motivation is clear, it is not fully demonstrated what is technically novel beyond applying existing generative models to this setting, and the paper would benefit from stronger ablations on prompt design / generation failure modes.

+ For Module 3 (Poison Selection), using ImageReward as an image-by-image filter is reasonable for improving realism, but it is unclear whether this reliably enforces trigger presence and backdoor-relevant quality (beyond general image-text alignment). In particular, the paper does not sufficiently quantify how much ImageReward-based selection improves ASR/Real ASR compared to simpler baselines (e.g., no filtering, CLIP similarity, object detection-based trigger verification), making it difficult to justify the necessity of this module.

**Requested Changes:**

See Weankness.

---

> ### Author Response · Authors · 2026-02-14
>
> Thank you for the thoughtful feedbacks.
>
> **Q1: The trigger suggestion module is not fully convincing as a necessary component. While the paper argues that VQA-based trigger recommendation reduces manual effort and improves stealth by selecting “compatible” objects, it is unclear whether this materially impacts attack effectiveness (e.g., Real ASR) beyond convenience. In practice, many backdoor triggers can work as long as poisoning and training are optimized, and the subsequent ImageReward-based filtering may already remove unrealistic samples even without a dedicated suggestion stage. The paper also lacks strong ablations comparing VQA-suggested triggers against simpler baselines (e.g., random trigger selection, manual selection, or heuristic choices based on visual consistency), making it hard to justify the added complexity of this module.**
>
> The reviewer’s concern is indeed reasonable; nevertheless, we’d like to clarify the role of trigger suggestion in reducing the manual effort.
>
> Trigger Suggestion reduces the effort (1) not only in selecting the trigger (of which the reviewer’s concerned), but (2) **also in determining its compatibility level w.r.t the dataset.** For (2), the trigger’s compatibility level cannot be easily determined without going through the entire dataset. Consequently, TriggerCraft’s suggestion module recommends the possible triggers and their compatibility levels, allowing the practitioners to choose whichever trigger to match their intended backdoor studies. Note that, there is usually a tradeoff between these attack effectiveness and compatability (as discussed in discussed in [1] and also in more details in our Q2’s response to Rev *iczt*).
>
> Nevertheless, our work also includes a similar baseline as suggested by the reviewer. We also include a human-suggested trigger (tennis ball – note that this baseline involves going through the entire dataset), acting as the control variable. Evidently, VQA-suggested triggers achieve similar performance as human-suggested triggers; hence, VQA-suggested triggers wouldn’t materially impact the ASR, echoing the reviewer’s statement of ``most backdoor triggers can work’’, including the suggested triggers.
>
>
> **Q2: The contributions of Module 2 (Trigger Generation) appear somewhat incremental, as the method mainly relies on standard diffusion-based image editing and text-to-image generation pipelines. While the motivation is clear, it is not fully demonstrated what is technically novel beyond applying existing generative models to this setting, and the paper would benefit from stronger ablations on prompt design / generation failure modes.**
>
> The uses of the existing generative models (in either the Generation module or other modules) are straightforward and indeed are not our contributions, as mentioned in the paper. Repurposing them to create backdoor datasets and providing the empirical analysis of their uses in the backdoor domain are our contributions. Specifically, the uses of generative models in the Generation Module are designed to be flexible, i.e., researchers can freely select prompts or generative models in order to study physical backdoors based on their interest. In the paper, we focus on its integration into and analysis within the whole framework, and researching advanced design of this component is definitely an interesting future extension.
>
> For example, as the reviewer suggests, a better prompt design is one potential direction. While we leave this for future exploration, we also consider the reviewer’s suggestion and change the lighting, sizes or angle of the physical triggers, i.e., introducing more diverse background prompts to the generative models, in order to bridge the gaps between CA and Real CA. As we can observe in the table below, a more diverse background helps minimize the CA and Real CA gaps.
>
>
> | Trigger|CA|ASR|Real CA|Real ASR|
> |---|---|---|---|---|
> |No Trigger|99.93|-|69.11|-|
> |Tennis Ball|99.63|89.23|65.75|86.86|
> |Book|99.73|97.13|68.20|44.25|
>
> **References:**
>
> [1] Wenger, E. et al. Backdoor attacks against deep learning systems in the physical world. CVPR’21

---

> > ### Author Response · Authors · 2026-02-14
> >
> > **Q3: For Module 3 (Poison Selection), using ImageReward as an image-by-image filter is reasonable for improving realism, but it is unclear whether this reliably enforces trigger presence and backdoor-relevant quality (beyond general image-text alignment). In particular, the paper does not sufficiently quantify how much ImageReward-based selection improves ASR/Real ASR compared to simpler baselines (e.g., no filtering, CLIP similarity, object detection-based trigger verification), making it difficult to justify the necessity of this module.**
> >
> > We’d like to note that in Fig. 12-15, we ablated the top and bottom images based on ImageReward’s scores, and notably, the images with low ImageReward scores are having implausible outputs (with missing or malformed triggers). Such images (with missing/malformed triggers) ideally should be excluded from the poisoned dataset, as it introduces noises to the backdoor learning.
> >
> > Nonetheless, we followed the reviewer’s suggestion and ran an ablation with and without our Poison Selection module. The result echoed our hypothesis, where the Real CA and Real ASR have significant gaps with the CA and ASR.
> >
> > |Setting|Trigger|CA|ASR|Real CA|Real ASR|
> > | --| --|--|--|--|--|
> > |With Poison Selection|Tennis Ball|99.63|89.23|65.75|86.86|
> > ||Book|99.73|97.13|68.20|44.25|
> > |Without Poison Selection|Tennis Ball|99.04|88.32|24.77|38.60|
> > ||Book|98.88|39.88|22.94|74.22|

---

### Decision · Action_Editor_5oo3 · 2026-03-03

**Recommendation:** Reject

**Audience:**

Yes

**Audience Explanation:**

The topic is of broad interest to TMLR

**Claims And Evidence:**

No

**Claims Explanation:**

The recommendation is based on the reviewers' comments, the action editor's evaluation, and the authors’ response.

This paper proposed a framework, TriggerCraft, as a new dataset for evaluating physical backdoor attacks. While the authors' rebuttal has addressed some of the reviewers' concerns, all reviewers share common concerns about the core claims. Specifically, the major issues and suggestions are listed below:

- Inconsistent Effectiveness and Questionable Scalability: The newly provided "Laptop" trigger experiment yields a Real ASR of only 35.93%. This significantly lower performance compared to simpler objects (like tennis balls) suggests that the framework's effectiveness does not generalize well to complex objects. The VQA-suggested "book" trigger fails to transfer robustly. The rebuttal shows its Real Attack Success Rate (Real ASR) drops to ~67% on ResNet-50 and DeiT-Small, contradicting claims that the synthetic datasets simulate real physical backdoors.

- Limited Evaluation Scope: The empirical evaluation remains restricted to a very small subset of ImageNet (only 5 classes). Also, the significant drop in clean accuracy calls into question the usefulness of the evaluation.Instead of providing a rigorous ablation study or feature analysis to explain why "Image Generation" attacks are exposed while "Image Editing" are not, the authors chose to remove the analysis entirely. This leaves a significant gap in our scientific understanding of the proposed method's security properties and its susceptibility to standard defenses.

- Marginal Advantage over Simple Baselines: While the authors demonstrated that TriggerCraft outperforms a "Cut-and-Paste" baseline in terms of Real ASR, the high complexity and computational cost of the three-module generative pipeline are not sufficiently justified by the resulting performance, especially given the low success rates on more complex, context-dependent triggers. One reviewer also made the comment that the framework introduced in this paper has almost identical naming to 3 stages introduced in Wang et al, which does not suggest a fundamental difference in the objective.


Overall, this submission should not be accepted in its current form. I hope the reviewers’ comments can help the authors prepare a better version of this submission.

**Resubmission Of Major Revision:**

The authors may consider submitting a major revision at a later time.